# Lipschitz regularized gradient flows and latent generative particles

## Abstract

Lipschitz regularized $f$-divergences are constructed by imposing a bound on the Lipschitz constant of the discriminator in the variational representation. These divergences interpolate between the Wasserstein metric and $f$-divergences and provide a flexible family of loss functions for non-absolutely continuous (e.g. empirical) distributions, possibly with heavy tails. We first construct Lipschitz regularized gradient flows on the space of probability measures based on these divergences. Examples of such gradient flows are Lipschitz regularized Fokker-Planck and porous medium partial differential equations (PDEs) for the Kullback-Leibler and $\alpha$-divergences, respectively. The regularization corresponds to imposing a Courant–Friedrichs–Lewy numerical stability condition on the PDEs. For empirical measures, the Lipschitz regularization on gradient flows induces a numerically stable transporter/discriminator particle algorithm, where the generative particles are transported along the gradient of the discriminator. The gradient structure leads to a regularized Fisher information which is the total kinetic energy of the particles and can be used to track the convergence of the algorithm. The Lipschitz regularized discriminator can be implemented via neural network spectral normalization and the particle algorithm generates approximate samples from possibly high-dimensional distributions known only from data. Notably, our particle algorithm can generate synthetic data even in small sample size regimes. A new data processing inequality for the regularized divergence allows us to combine our particle algorithm with representation learning, e.g. autoencoder architectures. The resulting particle algorithm in latent space yields markedly improved generative properties in terms of efficiency and quality of the synthetic samples. From a statistical mechanics perspective the encoding can be interpreted dynamically as learning a better mobility for the generative particles.

## 1 Introduction

We construct new algorithms that are capable of efficiently transporting arbitrary empirical distributions to a target data set. The transportation of the empirical distribution is constructed as a (discretized) gradient flow in probability space for Lipschitz-regularized $f$- divergences. Samples are viewed as particles and are transported along the gradient of the discriminator of the divergence towards the target data set. We take advantage of representation learning concepts, e.g. autoencoders, and make these algorithms efficient even in high-dimensional sample spaces by defining particle algorithms in latent space. Their accuracy is guaranteed by a new data processing inequality. One of our main tools is Lipschitz regularized $f$-divergences which interpolate between the Wasserstein metric and $f$-divergences. Such divergences Dupuis & Mao (2022); Birrell et al. (2022a;c), discussed in Section 2 provide a flexible family of loss functions for non-absolutely continuous distributions. In Machine Learning one needs to build algorithms to handle target distributions $Q$ which are singular, either by their intrinsic nature such as probability densities concentrated on low dimensional structures and/or because $Q$ is usually only known through $N$ samples (the corresponding empirical distribution $\widehat{Q}_N$ is always singular). Another key ingredient in our construction is that we build gradient flows where mass is transported along the gradient of the optimal discriminator in the variational formulation of the divergences. The time discretization of such gradient flows for empirical distributions gives rise to a so-called transporter/discriminator particle algorithm which transports an initial empirical distribution $\widehat{P}_N$ toward the target $\widehat{Q}_N$. The Lipschitz regularization

provides numerically stable, mesh free, particle algorithms that can act as generative models for high-dimensional target distributions. Moreover the gradient structure yields a dissipation functional which corresponds to the kinetic energy of the particles (a Lipschitz regularized version of Fisher information) and which can be used to control the convergence of the algorithm. The third new element in our methods is the use of representation learning to reduce the sample space dimension. We construct latent particle algorithms by building a Lipschitz regularized gradient flow in latent space. The fidelity of the latent space particle algorithm is guaranteed by a new data processing inequality for Lipschitz regularized divergence which ensures that convergence in latent space implies convergence in real sample space. The proposed generative approach is validated on a wide variety of datasets and applications ranging from image generation to gene expression data integration.

**Related work.** Our approach is inspired by the MMD and KALE gradient flows from Arbel et al. (2019); Glaser et al. (2021) based on an entropic regularization of the MMD metrics, and related work using the Kernelized Sobolev Discrepancy Mroueh et al. (2019). Furthermore, the recent work of Dupuis & Mao (2022); Birrell et al. (2022a) built the mathematical foundations for a large class of new divergences which contains the Lipschitz regularized $f$-divergences and used them to construct GANs, and in particular symmetry preserving GANs Birrell et al. (2022c)). Lipschitz regularizations (or related spectral normalization) have been shown to improve the stability of GANs Miyato et al. (2018); Arjovsky et al. (2017); Gulrajani et al. (2017). Our particle algorithms share similarities with GANs Goodfellow et al. (2014); Arjovsky et al. (2017), sharing the same discriminator but having a different generator step. They are also broadly related to the Wasserstein gradient flows Fan et al. (2022) which build a suitable neural method for the JKO-type schemes,Jordan et al. (1998). Furthermore, our methods are closely related to continuous time normalizing flows (NF) Chen et al. (2018a); Köhler et al. (2020); Chen et al. (2018b), diffusion models Sohl-Dickstein et al. (2015); Ho et al. (2020) and score-based generative flows Song & Ermon (2020); Song et al. (2021). However, the aforementioned continuous time models, along with variational autoencoders Kingma & Welling (2013) and energy based methods LeCun et al. (2006), are all likelihood-based. On the other hand, particle gradient flows such as the ones proposed here, can be classified in the same category of generative models that include GANs. Here there is more flexibility in selecting the loss function in terms of a suitable divergence or probability metric, enabling the comparison of even mutually singular distributions, e.g. Arjovsky et al. (2017). In Section A and Section F.1 we compare further our particle methods to other generative particles algorithms such as RKHS-based gradient flows and score-matching methods. Gradient flows in probability spaces related to the Kullback-Leibler (KL) divergence, such as the Fokker-Planck equations and Langevin dynamics Roberts & Tweedie (1996); Durmus & Moulines (2017) or Stein variational gradient descent Liu & Wang (2016); Liu (2017); Lu et al. (2019), form the basis of a variety of sampling algorithms when the target distribution $Q$ has a known density (up to normalization). The weighted porous media equations form another family of gradient flows based on $\alpha$-divergences Markowich & Villani (2000); Otto (2001); Ambrosio et al. (2005); Dolbeault et al. (2008); Vázquez (2014) which are very useful in the presence of heavy tails. Our gradient flows are Lipschitz-regularizations of such classical PDE's (Fokker-Planck and porous media equations), see Appendix B for a PDE and numerical analysis perspective on such flows. Finally, deterministic particle methods and associated probabilistic flows of ODEs such as the ones derived here for Lipschitz-regularized gradient flows for $(f, \Gamma)$ divergences, were considered in recent works for classical KL-divergences and associated Fokker-Planck equations as sampling tools Maoutsa et al. (2020); Boffi & Vanden-Eijnden (2022), for Bayesian inference Reich & Weissmann (2021) and as generative models Song et al. (2021). Our latent generative particles approach is inspired by latent diffusion models using auto-encoders Rombach et al. (2021) and by autoencoders used for model reduction in coarse-graining for molecular dynamics, Vlachas et al. (2022); Wang & Gómez-Bombarelli (2019); Stieffenhofer et al. (2021).

## 2 LIPSCHITZ-REGULARIZED $f$-DIVERGENCES

In the paper Dupuis & Mao (2022), continuing with Birrell et al. (2022a) a new general class of divergences has been constructed which interpolate between $f$-divergences and integral probability metrics and inherit desirable properties from both. In this paper we focus on one specific family which we view as a Lipschitz regularization of the KL-divergence (or $f$-divergences) or as an entropic regularization of the 1-Wasserstein metric. We denote by $\mathcal{P}(\mathbb{R}^d)$ the space of all Borel probability measures on $\mathbb{R}^d$ by $\mathcal{P}_1(\mathbb{R}^d) = \left\{ P \in \mathcal{P}(\mathbb{R}^d) : \int |x| dP(x) < \infty \right\}$. We denote by $C_b(\mathbb{R}^d)$ the

bounded continuous function and by $\Gamma_L = \{f : \mathbb{R}^d \to \mathbb{R} : |f(x) - f(y)| \leq L|x-y| \text{ for all } x, y\}$ the Lipschitz continous functions with Lipschitz constant bounded by $L$ (note that $a\Gamma_L = \Gamma_{aL}$).

**f-divergences.** If $f : [0, \infty) \to \mathbb{R}$ is strictly convex and lower-semicontinuous with $f(1) = 0$ the $f$-divergence of $P$ with respect to $Q$ is defined by $D_f(P\|Q) = E_Q[f(\frac{dP}{dQ})]$ if $P \ll Q$ and set to be $+\infty$ otherwise. We have the variational representation (see e.g. Birrell et al. (2022a) for a proof)

$$D_f(P\|Q) = \sup_{\phi \in C_b(\mathbb{R}^d)} \left\{ E_P[\phi] - \inf_{\nu \in \mathbb{R}} \{\nu + E_Q[f^*(\phi - \nu)]\} \right\} \tag{1}$$

where $f^*(s) = \sup_{t \in \mathbb{R}} \{st - f(t)\}$ is the Legendre-Fenchel transform of $f$. We will use the KL-divergence with $f_{\mathrm{KL}}(x) = x \log x$ and the $\alpha$-divergence: $f_\alpha = \frac{x^\alpha - 1}{\alpha(\alpha - 1)}$ with Legendre transforms $f^*_{\mathrm{KL}}(y) = e^{y-1}$ and $f^*_\alpha \propto y^{\frac{\alpha}{(\alpha-1)}}$ (see the Appendix). For KL the infimum over $\nu$ can be solved analytically and yields the Donsker-Varadhan with a $\log E_Q[e^\phi]$ term (see Birrell et al. (2022b) for more on variational representations).

**Wasserstein metrics.** The 1-Wasserstein metrics $W^{\Gamma_1}(P, Q)$ with transport cost $|x - y|$ is an integral probability metrics, see Arjovsky et al. (2017). By keeping the Lipschitz constant as a regularization parameter we set

$$W^{\Gamma_L}(P, Q) = \sup_{\phi \in \Gamma_L} \{E_P[\phi] - E_Q[\phi]\} \tag{2}$$

and note that we have $W^{\Gamma_L}(P, Q) = LW^{\Gamma_1}(P, Q)$.

**Lipschitz-regularized $f$-divergences.** The Lipschitz regularized $f$-divergences are defined directly in terms their variational representations, by replacing the optimization over bounded continuous functions in (1) by Lipschitz continuous functions in $\Gamma_L$.

$$D_f^{\Gamma_L}(P\|Q) := \sup_{\phi \in \Gamma_L} \left\{ E_P[\phi] - \inf_{\nu \in \mathbb{R}} \{\nu + E_Q[f^*(\phi - \nu)]\} \right\}. \tag{3}$$

Some of the important properties of Lipschitz regularized $f$-divergences, which summarizes results from Dupuis & Mao (2022); Birrell et al. (2022a) are given in Theorem 2.1. It is assumed there that $f$ is super-linear (called admissible in Birrell et al. (2022a)), that is $\lim_{s \to \infty} f(s)/s = +\infty$. This excludes the case of $\alpha$-divergences for $\alpha < 1$, for which the existence of optimizers is a more delicate problem, but parts of the theorems remain true.

**Theorem 2.1.** *Assume that $f$ is superlinear and strictly convex. Then for $P, Q \in \mathcal{P}_1(\mathbb{R}^d)$ we have*

*1. **Infimal Convolution Formula:** $D_f^{\Gamma_L}(P\|Q) = \inf_{\gamma \in \mathcal{P}(\Omega)} \left\{ W^{\Gamma_L}(P, \gamma) + D_f(\gamma\|Q) \right\}$ . In particular we have $0 \leq D_f^{\Gamma_L}(P\|Q) \leq \min \left\{ D_f(P\|Q), W^{\Gamma_L}(P, Q) \right\}$.*

*2. **Interpolation and limiting behavior of $D_f^{\Gamma_L}(P\|Q)$:***

$$\lim_{L \to \infty} D_f^{\Gamma_L}(P\|Q) = D_f(P\|Q) \quad \text{and} \quad \lim_{L \to 0} \frac{1}{L} D_f^{\Gamma_L}(P\|Q) = W^{\Gamma_1}(P, Q). \tag{4}$$

*3. **Optimizers:** There exists an optimizer $\phi^{L,*} \in \Gamma_L$, unique up to a constant in $\mathrm{supp}(P) \cup \mathrm{supp}(Q)$.*
*Remark* 2.2. The optimizer $\gamma^{L,*}$ in the infimal convolution formula exists, is unique and $d\gamma^{L,*} \propto (f^*)'(\phi^{L,*})dQ$ (see Birrell et al. (2022a) for details). For example for KL, $d\gamma^{L,*} \propto e^{\phi^{L,*}} dQ$.

## 3 LIPSCHITZ-REGULARIZED GRADIENT FLOWS AND THEIR TRANSPORTER/DISCRIMINATOR REPRESENTATION

**Background.** Gradient flows in latent spaces (for example in GANs) or in the space of probability measures (for example in diffusion models and score matching algorithm play a central role in

generative modeling. It is well-known Jordan et al. (1998) that the Fokker Planck equation (FPE) can be thought as the gradient flow of the KL divergence

$$\partial_t p_t = \text{div}\left(p_t \nabla \frac{\delta D_{KL}(p_t\|q)}{\delta p_t}\right) = \text{div}\left(p_t \nabla \log\left(\frac{p_t}{q}\right)\right) \tag{5}$$

where $p_t$ and $q$ are the densities at time $t$ and the stationary density respectively. A similar result relates weighted porous media equation and gradient flows for $f$ divergences Otto (2001). From a generative model perspective where $Q$ is known only through samples (and may not have a density in the first place as $Q$ is concentrated on low dimensional structure), one cannot use such flows without further regularization. Score matching and diffusion models start by regularizing the data by adding small amount noise to the data (see Sohl-Dickstein et al. (2015); Ho et al. (2020) andSong & Ermon (2020); Song et al. (2021)). Next, we propose a different and complementary approach by regularizing the divergence in (5) directly. We refer to Section A and Section F.1 for further connections between these different approaches and the last Example in Section 6.

**Lipschitz-regularized gradient flows.** Given a target probability measure $Q$, we build an evolution equation for probability measures based on the Lipschitz regularized $f$-divergences $D_f^{\Gamma_L}(P\|Q)$ by considering the PDE

$$\partial_t P_t = \text{div}\left(P_t \nabla \frac{\delta D_f^{\Gamma_L}(P_t\|Q)}{\delta P_t}\right), \quad P_0 = P \in \mathcal{P}_1(\mathbb{R}^d) \tag{6}$$

where $\frac{\delta D_f^{\Gamma_L}(P\|Q)}{\delta P}$ is the first variation of $D_f^{\Gamma_L}(P\|Q)$ (to be discussed below in Theorem 3.1). An advantage of the Lipschitz regularized $f$-divergences is its ability to compare singular measures and so (6) is to be understood in the sense of distributions (integrating against test functions). For this reason we use the probability measure $P_t$ notation in (6), instead of density notation $p_t$ as in the FPE (5). In the limit $L \to \infty$ and if $P \ll Q$, (6) yields the FPE (5) (for KL divergence) and the weighted porous medium equation (for $\alpha$-divergences) Otto (2001); Dolbeault et al. (2008), see Appendix B.

The following theorem was first proved in Dupuis & Mao (2022) for KL and can be generalized to the $f$-divergences considered in Theorem 2.1 (see the proof in Appendix C.

**Theorem 3.1.** *Assume $f$ is superlinear and strictly convex and $P, Q \in \mathcal{P}_1(\mathbb{R}^d)$. Then we have*

$$\frac{\delta D_f^{\Gamma_L}(P\|Q)}{\delta P}(P) = \phi^{L,*}. \tag{7}$$

*In more details, let $\rho$ be a signed measure of total mass 0 and let $\rho = \rho_+ - \rho_-$ where $\rho_\pm \in \mathcal{P}_1(\mathbb{R}^d)$ are mutually singular. If $P + \epsilon\rho \in \mathcal{P}_1(\mathbb{R}^d)$ for sufficiently small $|\epsilon|$ then $D_f^{\gamma_L}(P + \epsilon\rho\|Q)$ is differentiable at $\epsilon = 0$ and*

$$\lim_{\epsilon \to 0} \frac{1}{\epsilon}\left(D_f^{\Gamma_L}(P + \epsilon\rho\|Q) - D_f^{\Gamma_L}(P\|Q)\right) = \int \phi^{L,*} d\rho. \tag{8}$$

Combining Theorem 3.1 with (6) leads to a new class of PDEs:

**Transporter/Discriminator PDE:**

$$\partial_t P_t = \text{div}(P_t \nabla \phi_t^{L,*}), \quad \text{where} \quad \phi_t^{L,*} = \underset{\phi \in \Gamma_L}{\arg\max}\left\{E_{P_t}[\phi] - E_Q[f^*(\phi)]\right\} \tag{9}$$

*Remark* 3.2. **(a)** The transporter/discriminator PDE (9) makes sense when $P$ and $Q$ are replaced by their empirical measures $\hat{P}_N, \hat{Q}_N$ based on $N$ IID samples. This will be the basis of our numerical algorithm in Section 4 (see Algorithm 1). **(b)** Also (9) makes sense if $P$ and $Q$ are mutually singular (e.g. when $Q$ is supported on a low-dimensional structure). We can view (9) as a Lipschitz regularization of classical PDEs which allows particle-based approximations based on data. In particular, the Lipschitz condition on $\phi \in \Gamma_L$ enforces a finite speed of propagation of at most $L$ in the transport equation in (9). This is in sharp contrast with the Fokker Planck equation given in Appendix B which is a diffusion equation, see Appendix B.2 for more details and practical implications.

## 4 Lipschitz-regularized generative particles

In this section we build a numerical algorithm to solve the transporter/discriminator gradient flow when $N$ IID samples from the target distribution are given. For a map $T : \mathbb{R}^d \to \mathbb{R}^d$ and $P \in \mathcal{P}(\mathbb{R}^d)$, the pushforward measure is denoted by $T_\# P$ (i.e. $T_\# P(A) = P(T^{-1}(A))$). The forward-Euler discretization of the system (9) yields:

**Euler method for the Transporter/Discriminator PDE:**

$$P_{n+1} = \left(I - \Delta t \nabla \phi_n^{L,*}\right)_\# P_n, \quad \text{where} \quad \phi_n^{L,*} = \arg\max_{\phi \in \Gamma_L} \left\{ E_{P_n}[g] - E_Q[f^*(\phi)] \right\} \quad (10)$$

When only $N$ IID samples $\{X^{(i)}\}_{i=1}^N$ of the target distribution $Q$ are available we build a particle system by considering $N$ IID samples $\{Y^{(i)}\}_{i=1}^N$ from some initial measure $P$ ($M \neq N$ samples are also possible) and (10) becomes

**Lipschitz regularized generative particles:**

$$Y_{n+1}^{(i)} = Y_n^{(i)} - \Delta t \nabla \phi_n^{L,*}(Y_n^{(i)}), \quad \phi_n^{L,*} = \arg\max_{\phi \in \Gamma_L} \left\{ \frac{\sum_{i=1}^N \phi(Y_n^{(i)})}{N} - \frac{\sum_{i=1}^N f^*(\phi(X^{(i)}))}{N} \right\} \quad (11)$$

The empirical measure $\hat{P}_n^N = N^{-1} \sum_{i=1}^N \delta_{Y_n^{(i)}}$ built from (11) gives a solution of the system (10) if we use as target the empirical measure $\hat{Q}^N = N^{-1} \sum_{i=1}^N \delta_{X^{(i)}}$ and as initial condition the empirical measure $\hat{P}^N = N^{-1} \sum_{i=1}^N \delta_{Y_0^{(i)}}$. Finally we note that (11) is a time-discretization of the Lagrangian formulation of (9), i.e. the ODE/variational problem

$$\frac{d}{dt} Y_t = -\nabla \phi^{L,*}(Y_t, t), \quad \text{where} \quad \phi^{L,*} = \arg\max_{\phi \in \Gamma_L} \left\{ E_{P_t}[\phi] - \inf_{\nu \in \mathbb{R}} \left\{ \nu + E_Q[f^*(\phi - \nu)] \right\} \right\}. \quad (12)$$

---

**Algorithm 1:** Lipschitz regularized generative particles algorithm

**Require:** $f$ defined in (2) and its Legendre conjugate $f^*$, $L$: Lipschitz constant, $\nu$: scalar parameter for optimizing $f$ divergence,

1   $T$: number of updates for the particles, $\gamma$: time step size, $N$: number of particles

    **Require:** $W = \{W^l\}_{l=1}^D$: parameters for the neural network $\phi : \mathbb{R}^d \to \mathbb{R}$, $D$: depth of the neural network, $\delta$: learning rate of the neural network, $T_{\text{NN}}$: number of updates for the neural network.

**Result:** $\{Y_T^{(i)}\}_{i=1}^N$

2   Sample $\{X^{(i)}\}_{i=1}^N \sim Q$, a batch from the real data

3   Sample $\{Y_0^{(i)}\}_{i=1}^N \sim P_0 = P$, a batch of prior samples

4   Initialize $\nu \leftarrow 0$, $W$ randomly and $W^l \leftarrow L^{1/D} * W^l / \|W^l\|_2$

5   **for** $n = 0$ **to** $(T-1)$ **do**

6     **for** $m = 0$ **to** $T_{\text{NN}} - 1$ **do**

7        $grad_{W,\nu} \leftarrow \nabla_W \left[ N^{-1} \sum_{i=1}^N \phi(Y_n^{(i)}; W) - \left\{ N^{-1} f^* \left( \phi(X_n^{(i)}; W) - \nu \right) + \nu \right\} \right]$

8        $W \leftarrow W + \delta * grad_W, \quad \nu \leftarrow \nu + \delta * grad_\nu$

9        $W^l \leftarrow L^{1/D} * W^l / \|W^l\|_2$

10    **end**

11    $Y_{n+1}^{(i)} \leftarrow Y_n^{(i)} - \gamma \nabla \phi_n^L(Y_n^{(i)}; W), \quad i = 1, \cdots, N$

12   **end**

```
// The height of φ is adjusted by the optimization over ν.
   This keeps φ(X_n^(i); W) values to reside in the domain of f*  and
   allows to avoid the degeneracy.
```

---

*Remark* 4.1. **(a)** The transport mechanism given by (11) is linear. However, nonlinear interactions between particles as introduced via the discriminator $\hat{\phi}_n^{L,*}$ are created due to the velocity field $\nabla \phi_n^{L,*}$ which depends on all particles that comprise $\hat{P}_n^N$ and $\hat{Q}^N$ at each step $n$. **(b)** Computationally, the

discriminator optimization (over Lipschitz continuous functions) is implemented, for example, via spectral normalization for neural networks architectures. Moreover the gradient of the discriminator is computed only at the positions of the particles. **(c)** The Lipschitz bound $L$ on the discriminator space implies a pointwise bound $|\nabla\phi_n^{L,*}(Y_n^{(i)})| \leq L$ and thus the particle speed is bounded by $L$. Hence the Lipschitz regularization imposes a speed limit $L$ on the particles, ensuring the stability of the algorithm for suitable choices of $L$. This implicit grid is reminiscent of the Courant, Friedrichs, and Lewy (CFL) condition for the stability of discrete scheme. These are fundamental features for the performance and the stability of Algorithm 1 derived from (11) (see Sections 6) and Appendix B.

**Kinetic energy of particles:** The gradient structures implies, see Theorem B.1, that, for (9), the derivative of the regularized divergence satisfies $\frac{d}{dt}D_f^{\Gamma_L}(P_t\|Q) = -I_f^{\Gamma_L}(P_t\|Q)$ where $I_f^{\Gamma_L}(P_t\|Q) = E_{P_t}[|\nabla\phi^{L,*}|^2]$ which is interpreted as a Lipschitz-regularized Fisher Information. As $L \to \infty$ one recovers for example the Fisher Information used for the Fokker-Planck equation. For the Algorithm 1 the Lipshitz-regularized Fisher information

$$I_f^{\Gamma_L}(\hat{P}_n^N\|\hat{Q}^N) = \int |\nabla\phi_n^{L,*}|^2 \hat{P}_n^N(dx) = \frac{1}{N}\sum_{i=1}^{N}|\nabla\phi_n^{L,*}(Y_n^{(i)})|^2,\tag{13}$$

is equal to the total kinetic energy of the particle since $\nabla\phi_n^{L,*}(Y_n^{(i)})$ is the velocity of the $i^{th}$ particle at time $n$. Clearly when the total kinetic energy $I_f^{\Gamma_L}(\hat{P}_n^N\|\hat{Q}^N)$ is zero, the algorithm will stop.

## 5 LATENT GENERATIVE PARTICLES: GRADIENT FLOWS IN LATENT SPACE

A standard paradigm of machine learning is that target measures are often supported on low dimensional structures. We leverage this insight, in the form of an auto-encoder, to construct particle algorithms in a latent, lower dimensional space. The resulting latent particle algorithms are both more accurate and efficient, even in high-dimensional sample spaces, and their performance is guaranteed by a new Data Processing Inequality in Theorem 5.1. Assume $Q = Q^{\mathcal{Y}}$ is supported on some low dimensional set $S \subset \mathcal{Y} = \mathbb{R}^d$, an encoder map $\mathcal{E} : \mathcal{Y} \to \mathcal{Z}$ where $\mathcal{Z} \subset \mathbb{R}^{d'}$, $d' < d$ and a decoder map $\mathcal{D} : \mathcal{Z} \to \mathcal{Y}$ are invertible in $S$, i.e. $\mathcal{D} \circ \mathcal{E}(S) = \mathcal{D}(\mathcal{Z}) = S$. We denote by $\mathcal{E}_{\#}Q^{\mathcal{Y}}$ the image of the measure $Q^{\mathcal{Y}}$ by the map $\mathcal{E}$, i.e. for $A \subset \mathcal{Z}$, we define $\mathcal{E}_{\#}Q^{\mathcal{Y}}(A) = Q^{\mathcal{Y}}(\mathcal{E}^{-1}(A))$ and likewise for $\mathcal{D}_{\#}P^{\mathcal{Z}}$. The following theorem expresses how information remains controlled under encoding/decoding and guarantees the performance of the approximation $\mathcal{D}_{\#}P^{\mathcal{Z}}$ in the real space. The latter is achieved by an *a posteriori* estimate (14), in the sense of numerical analysis, where the approximation in the tractable latent space $\mathcal{Z}$ will bound the error in the real space $\mathcal{Y}$.

**Theorem 5.1.** *Suppose that*

1. **Perfect encoding.** *For $Q^{\mathcal{Y}}$ the encoder $\mathcal{E}$ and the decoder $\mathcal{D}$ are such that $\mathcal{D}_{\#}\mathcal{E}_{\#}Q^{\mathcal{Y}} = Q^{\mathcal{Y}}$.*

2. **Lipschitz decoder.** *The decoder is Lipschitz continuous with Lipschitz constant $a_{\mathcal{D}}$.*

*Then, for any $P^{\mathcal{Z}} \in \mathcal{P}_1(\mathcal{Z})$ we have*

$$D_f^{\Gamma_L}\left(\mathcal{D}_{\#}P^{\mathcal{Z}}\|Q^{\mathcal{Y}}\right) \leq D_f^{\Gamma_{a_{\mathcal{D}}L}}\left(P^{\mathcal{Z}}\|\mathcal{E}_{\#}Q^{\mathcal{Y}}\right).\tag{14}$$

This theorem and more general versions thereof for other representation learning tools beyond autoencoders, is proved in Appendix D.2. The proof is a consequence of a new, tighter data processing inequality derived in Birrell et al. (2022a) that involves both transformations of probabilities and discriminator spaces $\Gamma$.

*Remark* 5.2. In practice an autoencoder is trained on data using the empirical measure $\hat{Q}_N$ and suitable loss function and neural network architectures. Assumption 2 in Theorem 5.1 can easily be enforced using e.g. spectral normalization. Assumption 1 is a reasonable, but somewhat idealized, version of the requirement that the autoencoder captures adequately the features of the dataset $Q$. In particular the dimension of the latent space $\mathcal{Z}$ needs to be selected carefully (see Section 6).

**Gradient flow in latent spaces.** If $\phi_t^{\mathcal{Z}}$ is the discriminator in latent space leading to the gradient flow (9), $\partial_t P_t^{\mathcal{Z}} = \mathrm{div}(P_t^{\mathcal{Z}} \nabla \phi_t^{\mathcal{Z}})$ then, in the particle algorithm, each particle is transported following (the time-discretization of) the ODE $\dot{z}_t = -\nabla \phi_t^{\mathcal{Z}}(z_t)$, as in Section 4. The Algorithm 2 can be found in the Appendix D.3. Upon decoding we find the transport ODE in real space is

$$\dot{y}_t = \left(\frac{\partial \mathcal{D}}{\partial z}(z_t)\right)^T \dot{z}_t = -\left(\frac{\partial \mathcal{D}}{\partial z}(z_t)\right)^T \frac{\partial \mathcal{D}}{\partial z}(z_t) \nabla_y \phi_t^{\mathcal{Y}}(\mathcal{D}(z_t)) \tag{15}$$

where $\frac{\partial \mathcal{D}}{\partial z}(z_t)$ is the Jacobian of $\mathcal{D}$ at the point $z_t$ and the reconstructed discriminator $\phi^{\mathcal{Y}}$ is given by $\phi^{\mathcal{Z}} = \phi^{\mathcal{Y}} \circ \mathcal{D}$. Using (15) we can therefore interpret the encoding as learning a mobility $\mu_t = \frac{\partial \mathcal{D}}{\partial z}(z_t)^T \frac{\partial \mathcal{D}}{\partial z}(z_t)$, i.e., learning a better geometry in real space. This leads to a gradient flow in real space with non-trivial mobility, cf. (9),

$$\partial_t P_t^{\mathcal{Y}} = \mathrm{div}\left(\mu_t P_t^{\mathcal{Y}} \nabla \phi_t^{\mathcal{Y}}\right) . \tag{16}$$

We note that the mobility concept is well-known in computational materials science where it is used to model kinetics of species and interfaces, see for instance Cahn (1965); Zhu et al. (1999); Wang et al. (2020). Finally, we note a similar computation to (15) in Mroueh et al. (2019) regarding the interpretation of GAN's as a gradient flow. The differences (and similarities) between (Lipschitz-regularized) Generative particle algorithm (GPA) and GAN are summarized in Figure 4 and Table 1, where in the latter we also include a comparison between mobilities.

| | GPA | GPA in a latent space | GAN |
|---|---|---|---|
| Discriminator | $\phi^{\mathcal{Y}} \in \mathrm{Lip}(\mathcal{Y})$ | $\phi^{\mathcal{Z}} \in \mathrm{Lip}(\mathcal{Z})$ | $\phi^{\mathcal{Y}} \in \mathrm{Lip}(\mathcal{Y})$ |
| Generator | $(I_{\mathcal{Y}} - \Delta t \nabla \phi^{\mathcal{Y}})_{\#} P_n^{\mathcal{Y}}$ | $\left(\mathcal{D} \circ (I_{\mathcal{Z}} - \Delta t \nabla \phi^{\mathcal{Z}})\right)_{\#} P_n^{\mathcal{Z}}$ | $\mathcal{G}_\theta(z), z \sim \mathcal{N}(\mathbf{0}, I_{\mathcal{Z}})$ |
| Updates | Particles $y \in \mathcal{Y}$ | Particles $z \in \mathcal{Z}$ | Generator parameters $\theta \in \mathbb{R}^{|\theta|}$ |
| Mobility $\mu$ (16) | $I_{\mathcal{Y}}$ | $\frac{\partial \mathcal{D}}{\partial z}(z_t)^T \frac{\partial \mathcal{D}}{\partial z}(z_t)$ | $\frac{\partial \mathcal{G}(\theta_t, z)}{\partial \theta}^T \frac{\partial \mathcal{G}(\theta_t, z)}{\partial \theta}$ |

Table 1: Comparison of features in GPA, GPA in the latent space and GAN. Note that GPA consists of one neural network for the discriminator while GAN has two neural networks for the discriminator and the generator, respectively. We use the notations $\mathcal{Y} = \mathbb{R}^d$ and $\mathcal{Z} = \mathcal{E}(\mathcal{Y})$ for distinguishing the real space and the latent space. A schematic diagram in Figure 4 shows how the Latent GPA and the GAN interact between the latent and the real spaces.

## 6 EXPERIMENTS

We present three types of experiments: (1) generating MNIST images from a small number of data, see Figure 1; (2) merging gene expression data sets which are very high-dimensional (and thus require a latent space description) but also have a low number samples on the order of the low hundreds since they correspond to patients, see Figure 2; (c) generating heavy-tailed distributions where KL or maximum likelihood-based methodologies will necessarily fail since an $f$-divergence is required, see Figure 3. Overall, we found that GPA perform best in learning from small number of samples, learning distributions with heavy tails, and exhibit enhanced learning when a latent space is available.

**1. Learning MNIST from a few data.** Our $(f_{\mathrm{KL}}, \Gamma_1)$-GPA is found to perform well in generating images using a small number of samples while GANs struggle with limited data. We stress from the digit-conditional MNIST data generation example in Figure 1 that $(f_{\mathrm{KL}}, \Gamma_1)$-GPA could generate images from ten digit labels out of 200 samples. We compare the performance with $(f_{\mathrm{KL}}, \Gamma_1)$-GAN and the well-known Wasserstein GAN. FID values and the improvement of our result with latent GPA can be found in Appendix F.2.

**2. Merging of microarray gene expression data sets** Using GPA, we can transport arbitrary source data to arbitrary target data even if a relatively small number of samples is available from the target. Furthermore, combining our algorithm with representation learning, we introduce a bioinformatics application of our algorithm in the analysis of gene expression data. Gene expression datasets

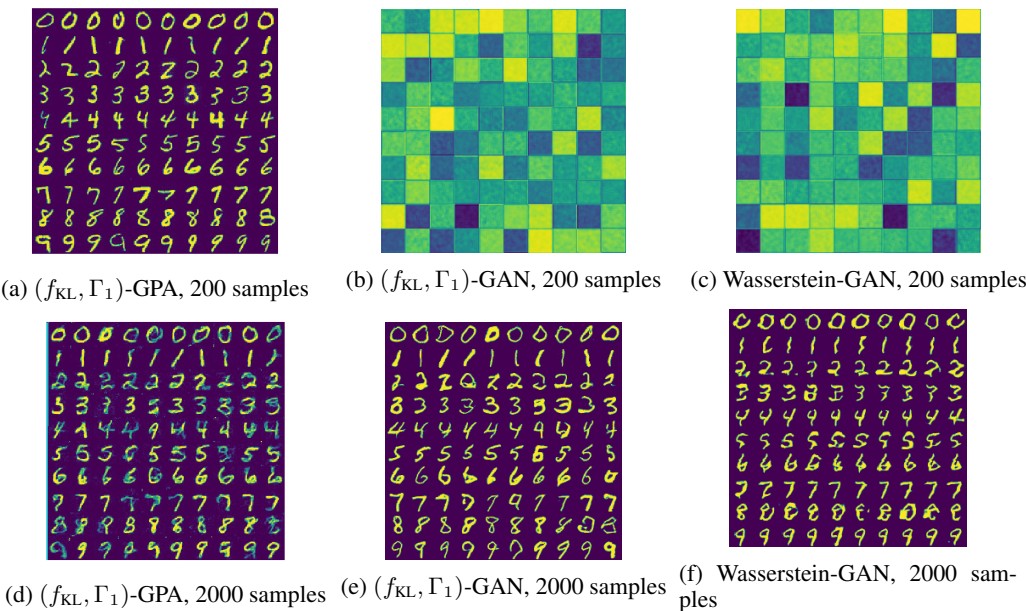

(a) $(f_{\mathrm{KL}}, \Gamma_1)$-GPA, 200 samples

(b) $(f_{\mathrm{KL}}, \Gamma_1)$-GAN, 200 samples

(c) Wasserstein-GAN, 200 samples

(d) $(f_{\mathrm{KL}}, \Gamma_1)$-GPA, 2000 samples

(e) $(f_{\mathrm{KL}}, \Gamma_1)$-GAN, 2000 samples

(f) Wasserstein-GAN, 2000 samples

Figure 1: **MNIST; Learned digits by different generative models (column) with different number of training data (row).** The $(f_{\mathrm{KL}}, \Gamma_1)$-GPA was able to learn digits from a small data set, while the other methods failed. Using sufficiently large training data, GANs outperformed in capturing the scale, which can be observed by the more intense color contrast between a digit and its background. See FID scores in Table 6.

are not only high-dimensional but also small-sized thus it is crucial to increase the sample size by integrating together all available datasets from the same disease. However, this is not a straightforward process since it is well known that gene expression datasets may have different statistics even when they target the same disease; a phenomenon referred to as "batch effects" Tran et al. (2020). We propose to mitigate batch effects via the latent generative particle algorithm and match the statistics of the two datasets. Figure 2 presents the results on applying our algorithm between two breast cancer datasets from the Gene Expression Omnibus (https://www.ncbi.nlm.nih.gov/geo/). More technical details and experiments can be found in the Appendix F.3.

**3. GPA and Porous medium equations for heavy-tailed targets**   Similarly to score-based methods (see equation 22 in Section A) developed in Boffi & Vanden-Eijnden (2022) to solve the FP equation 5, our proposed methods will allow to develop new particle systems algorithms for solving porous medium equations with steady state probability measure $Q$ and density $q$:

$$\partial_t p_t = \mathrm{div}\left(p_t \nabla f'\left(\frac{p_t}{q}\right)\right) \qquad (17)$$

For the special case where the $f$-divergence is an $\alpha$-divergence with $f_\alpha(x) \sim x^\alpha$, we obtain a power law in equation 17 and ultimately the well-known porous media equations, e.g. Bonforte et al. (2010); Otto (2001); Dolbeault et al. (2008), used for applications to actual porous medium flow, typically in dimension 3. However, here we propose porous medium equations and associated particle algorithms as *statistical learning tools* for pdfs with heavy tails. For instance, score-based methods, are KL-based, see Song et al. and the discussion in Section A, hence they are not suitable for heavy tailed distributions: we refer to the collapse in the algorithms that minimize KL divergence observed in Figure 2 in Birrell et al. (2022a), in stark contrast to the stable behavior of suitable $(f, \Gamma)$-divergences for heavy tails. For this reason, we propose GPA algorithms based on the porous medium equation equation 17 and its more stable Lipschitz-regularized particles-based solution given by the Algorithm 1 as a new efficient and reliable way for generating samples from heavy-tailed targets $Q$. See the example in Figure 3. Our methods are also partly theoretically grounded on of asymptotics and related functional inequalities results for porous medium equations, e.g. Dolbeault et al. (2008); Bonforte et al. (2010).

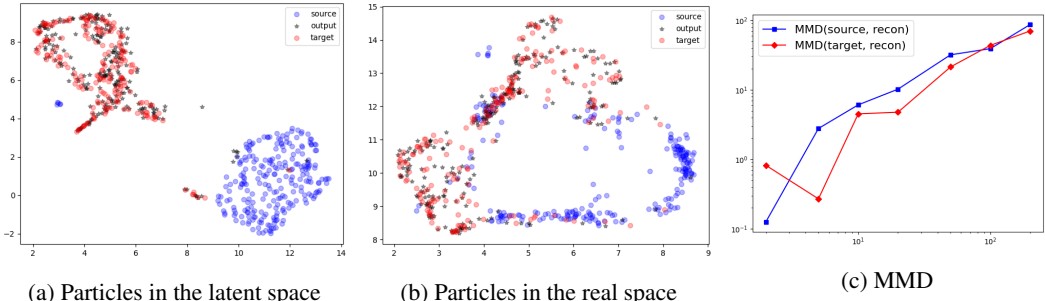

(a) Particles in the latent space      (b) Particles in the real space      (c) MMD

Figure 2: **(Gene expression data; Merging Breast Cancer datasets).** We merged gene data using our latent GPA in significantly lower dimensions. Two distinct gene data sets but from the same disease decrease their dimension from 54,675D into $d' = 2, 5, 10, 20, 50, 100, 200$ using normalized PCA. Then, latent particles are transported using GPA. **blue: source (206 samples), red: target (245 samples), black: transported (206 samples)**. **(a)** Latent particles in the $\mathbb{R}^{d'}$ with $d' = 20$ which are encoded by the PCA. **(b)** Transported samples are reconstructed to the real space. The 2D visualizations are obtained using the UMAP algorithm McInnes et al. (2018). **(c)** The MMD distance Gretton et al. (2012) between the reconstructed datasets. **blue: $\mathbf{MMD}(P_0^{\mathcal{Y}}, P_T^{\mathcal{Y}})$, red: $\mathbf{MMD}(Q^{\mathcal{Y}}, P_T^{\mathcal{Y}})$**, $T = 25K$. The transported distribution has smaller distances from the target distribution when $d' = 5, 10, 20, 50, 200$.

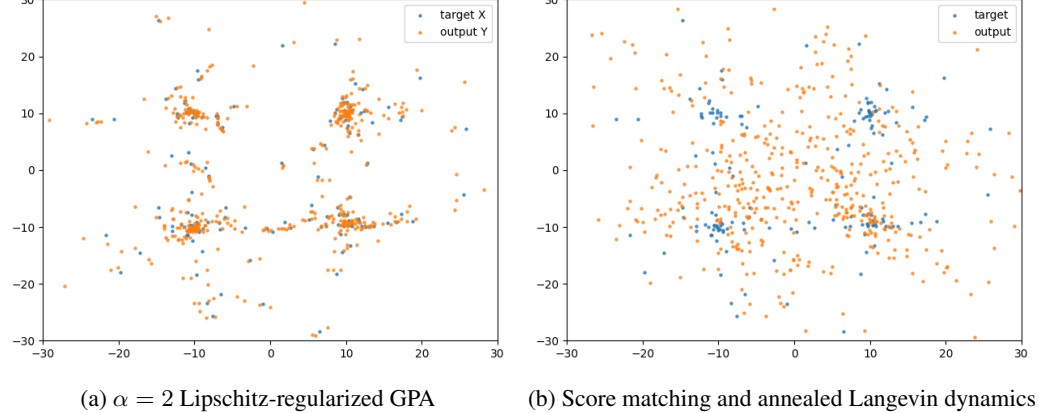

(a) $\alpha = 2$ Lipschitz-regularized GPA      (b) Score matching and annealed Langevin dynamics

Figure 3: **(2D Mixture of Student-t)** $(f_\alpha, \Gamma_1)$**-GPA with $\alpha = 2$ for a heavy tailed target and comparison with score based model.** 200 target samples from Student-t($\nu$) with $\nu = 0.5$ are provided to transport 500 particles which are uniformly distributed in the plotted region at time $t = 0$. Blue: target, Orange: output. **(a)** The choice of divergence $f_\alpha$ with $\alpha = 2$ and propagation of particles through the $(f_\alpha, \Gamma_1)$-GPA captures the heavy-tailed target. **(b)** Noise conditional score network (NCSN, score based model Song et al.) evolves particles by learning the vector field, the score $\nabla \log Q(x)$, from data. However, a mixture of disjoint distributions makes it hard to learn the score where the data is sparse. NCSN tackles the problem by injecting different levels of noise on the data and learning the scores of noise-injected distributions. Then it propagates particles through annealed Langevin dynamics using a sequence of scores $s_\sigma$ with different noise levels $\sigma$. When the level of injected noise gets smaller as much as $\sigma \leq 1$, score-matching of the perturbed data for Student-t was extremely hard. In addition, particles transportation through (annealed) Langevin dynamics might not lead to the convergence to the heavy tailed distribution. A similar comparison for a mixture of Gaussians can be found in Figure 6.

## 7 REPRODUCIBILITY STATEMENT

Each figure contains main parameters such as divergence-specifying $f$, Lipschitz constant $L$, and dataset parameter. In Appendix Section E, experimental setting for the experiments in the main text and appendix are described in these aspects:

- Data sets
- Neural network architectures
- Computational resources.

In Supplementary material, source code, all dependent libraries, and documentation (README.md) are attached. README.md specifies the required open-source libraries and the entire parameters set including random seeds for reproducing individual experiments.

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

## A  CONNECTIONS OF GPA WITH SCORE GENERATIVE METHODS

In this Section we discuss connections of GPAs with diffusion-type generative models in particular with score-based models which seem to be the most closely related. Score-based generative modeling (SGM) relies closely on concepts and methods related to Langevin samplers, e.g. Durmus & Moulines (2017). Given $\mathbb{R}^d$–valued samples $\{Y^{(i)}\}_{i=1}^N$ from an unknown probability $Q$ with distribution $q$, we want to produce more realizations from $Q$. In SGMs, the score of the unknown distribution, $\nabla \log q(y)$ is learned from the training set. An optimization problem to learn the score is defined as follows Song & Ermon (2020): we search for $s : \mathbb{R}^d \rightarrow \mathbb{R}^d$ in a function space $\mathcal{F}$ parametrized by $\theta$ (typically neural networks),

$$\min_\theta L(\theta) = \min_\theta \frac{1}{2} \int_{\mathbb{R}^d} \|s(y; \theta) - \nabla \log q(y)\|^2 q(y) dy. \tag{18}$$

The key observation in (18) is that the loss functional can be estimated via available samples from $Q$ without any evaluation of the density $q$ (of $Q$) or $\nabla \log q$, Hyvärinen (2005). Indeed, by expanding the square and integrating by parts in (18) we arrive at an equivalent loss function $J(\theta)$:

$$\text{argmin}_\theta L(\theta) = \text{argmin}_\theta L_0(\theta) = \text{argmin}_\theta E_Q \left[ \frac{1}{2} \|s(y; \theta)\|_2^2 + \nabla_y \cdot s(y; \theta) \right]. \tag{19}$$

Since computing the divergence of $s(x; \theta)$, when it is expressed as some neural network is quite expensive, the complexity of this last loss functional (19) can be further reduced by using a Hutchinson-type randomization Hutchinson (1989) for the efficient evaluation of $\nabla_y \cdot s(y; \theta)$, by developing the so-called "sliced score matching" method Song et al. (2020). Lastly, we remark that it may initially appear that the selection of the loss functional (18) is somewhat ad hoc as in many other successful machine learning algorithms. However, using the Girsanov Theorem it can be shown that (18) has rigorous information-theoretic underpinnings because it is related to the minimization of the KL divergence on path-space, Song et al..

After learning the score from the minimization of (19), namely $\hat{s}(y) = s(y; \theta^*) \approx \nabla \log q(y)$, there are several directions that can be taken towards generative modeling. The most naïve option is to simulate trajectories of the overdamped Langevin dynamics using the learned score:

$$dY_t = \hat{s}(Y_t)dt + \sqrt{2}dW_t, \quad \text{where} \quad \hat{s}(y) \approx \nabla \log q(y). \tag{20}$$

As discussed in Song & Ermon (2020), naïvely applying this approach is not practical: (a) Langevin dynamics will not sample from the true data distribution when the data lies on a lower dimensional manifold; (b) the score cannot be accurately estimated for regions with little data; (c) Langevin dynamics mixes poorly, and therefore generates poorly, especially for multi-modal systems. To overcome these challenges, Song & Ermon (2020); Song et al. (2021) propose to add noise to the available data from $Q$ in a systematic fashion. In Song & Ermon (2020), the authors propose to build a noise-conditional score network, in which a sequence of noisy datasets are generated by adding different noises to the given dataset. In the end, we simulate an annealed Langevin dynamics analogue to (20):

$$dY_t = \sigma^2(t)\nabla \log q(Y_t)dt + \sqrt{2}\sigma(t)dW_t, \tag{21}$$

with the annealing schedule $\sigma(t)$. The distribution of $Y_t$ at some finite cut-off time $T$ is supposed to approximate the data distribution $q$. Indeed (21) is shown to mix better, and estimate the score better in regions of low probability, Song & Ermon (2020). In Song et al. (2021), the author propose an alternative method by considering a forward-backward formulation. However all these Langevin dynamics approaches require tuning — mainly since the annealing schedule or the forward model need to be user-prescribed and is problem-dependent, like all annealing methods.

**Connections between Score-based methods and GPA.**  The distribution $p_t$ of the solution $Y_t$ of the SGM (20) is the solution of classical Fokker Planck (FP) equation (5), given a target distribution $q$. The FP is of course a diffusion equation, but it can also be viewed as a transport equation taking the form $\partial_t p_t = \nabla \cdot (v(x_t, t)p_t)$, with velocity field $v(x, t) = \nabla \frac{\delta KL(p_t \| q)}{\delta p_t}$. In analogy to (12) we can write the solution of this transport formulation as the density of particles evolving according to its Lagrangian formulation

$$\frac{d}{dt}Y_t = v(Y_t, t) = \nabla \log q(Y_t) - \nabla \log p_t(Y_t), \quad \text{where } Y_t \sim P_t. \tag{22}$$

In Song et al. (2021), the authors proposed the deterministic probability flow (22) as an alternative to generative stochastic samplers such as (20) and (21) due to advantages related to better statistical estimators.

In the perspective of GPAs, we can write (22), in an equivalent transport/variational problem that employs the variational formulation of the KL divergence, (1). First, we note that $\phi^*(x) = \log \frac{p_t(x)}{q(x)}$ is the maximizer (discriminator) of this variatonal representation, Birrell et al. (2022b). Next, notice that (22) can also be written in the equivalent particle/variational form (12) which for the KL divergence, and due to $\phi^*(x) = \log \frac{p_t(x)}{q(x)}$, takes the form

$$\frac{d}{dt}X_t = v(X, t) = -\nabla \phi^*(X, t), \quad \text{where} \quad \phi^* = \text{argmax}\left\{E_{P_t}[\phi] - \inf_{\nu \in \mathbb{R}}\left\{\nu + E_Q[e^{\phi - \nu - 1}]\right\}\right\}. \tag{23}$$

It is evident that (23) is mathematically equivalent to (22), since $\phi^*(x) = \log \frac{p_t(x)}{q(x)}$. However, the particle/variational problem (23)–through its generalization (12)–allows us to use a much wider variety of divergences that–unlike KL used in score-based methods–are suitable for probability distributions supported at lower dimensional manifolds, are singular such as empirical distributions, or have heavy tails. These are precisely the featues of the GPA developed in in this paper, based on $(f, \Gamma)$-divergences.

## B  PDE, CONVERGENCE & LEARNING RATES

Here, we discuss how PDE perspectives and tools provide insights for the analysis, stability and convergence for the proposed generative particle algorithms. We focus on continuous time for convenience. By recalling Theorem 2.1, part 4. and $\gamma^{L,*} \to P$ as $L \to \infty$ (if $P$ is absolutely continuous with respect to $Q$) in Remark 2.2, (9) becomes a Lipschitz-regularized $f$-divergences gradient flow (with its limit as $L \to \infty$), i.e.

$$\underbrace{\partial_t P_t = \text{div}\left(P_t \nabla f'\left(\frac{d\gamma^{L,*}}{dQ}\right)\right)}_{\text{Lip. regularized } f\text{-divergence flow}} \quad \underset{L \to \infty}{\longrightarrow} \quad \underbrace{\partial_t P_t = \text{div}\left(P_t \nabla f'\left(\frac{dP_t}{dQ}\right)\right)}_{f\text{-divergence flow}} \tag{24}$$

The right hand side of (24) is a nonlinear operator which encodes the Lipschitz regularization in the discriminator space. This defines a new class of PDE gradient flows where absolute continuity between $P_t$ and $Q$ for every $t \geq 0$, is not required, contrary to gradient flows of $f$-divergences (obtained as $L \to \infty$.) We discuss these connections next in the context of two well-known PDEs, the (linear) Fokker-Planck and the (non-linear) porous media equation. Rewriting the limiting equation in terms of the density $h_t = \frac{dP_t}{dQ}$ we have

**1. Lipschitz-regularized Fokker-Planck.**  For the KL, $f(x) = x \log(x)$ we obtain

$$\partial_t P_t = \text{div}\left(P_t \nabla \log\left(\frac{d\gamma_t^{\Gamma_L,*}}{dQ}\right)\right) \quad \underset{L \to \infty}{\longrightarrow} \quad \partial_t h_t = (\Delta + \nabla \log(q) \cdot \nabla)h_t \tag{25}$$

**2. Lipschitz-regularized Weighted Porous Medium equation (WPME).** For the $\alpha$-divergence with $f_\alpha(x) = \frac{1}{\alpha(\alpha-1)}x^\alpha$ we obtain a regularization of the porous media equation Otto (2001); Dolbeault et al. (2008)

$$\partial_t p_t = \frac{1}{\alpha - 1}\text{div}\left(p_t \nabla \left(\frac{\eta_t^{\Gamma_L,*}}{q}\right)^{\alpha-1}\right) \quad \underset{L \to \infty}{\longrightarrow} \quad \partial_t h_t = \frac{1}{\alpha}\left(\Delta + \nabla \log q \cdot \nabla\right) h_t^\alpha \tag{26}$$

### B.1  CONVERGENCE TO EQUILIBRIUM AND FUNCTIONAL INEQUALITIES

Functional inequalities are fundamental methods for guaranteeing the convergence of gradient flow PDE to their equilibrium states and therefore are natural tools for studying convergence properties for the corresponding particle-based algorithms. A first step is to compute the rate of change of the divergence along solutions $P_t$ of (9).

**Theorem B.1.** [Lipschitz regularized dissipation] *Along a trajectory of a smooth solution $\{P_t\}_{t\geq 0}$ of (9) with source probability distribution $P$ we have the following rate of decay identity:*

$$\frac{d}{dt} D_f^{\Gamma_L}(P_t\|Q) = -I_f^{\Gamma_L}(P_t\|Q) \leq 0 \tag{27}$$

*where we define the Lipschitz-regularized Fisher Information as*

$$I_f^{\Gamma_L}(P\|Q) = \int |\nabla \phi^{L,*}|^2 P(dx) = E_P\left[\left|\nabla f'\left(\frac{d\gamma^{L,*}}{dQ}\right)\right|^2\right]. \tag{28}$$

*Consequently, for any $T \geq 0$, we have $D_f^{\Gamma_L}(P_T\|Q) = D_f^{\Gamma_L}(P\|Q) - \int_0^T I_f^{\Gamma_L}(P_s\|Q)ds$.*

*Remark* B.2. **(a)** For the generative particles the Lipschitz-regularized Fisher Information can be interpreted as their total kinetic energy, see Paragraph 12. **(b)** When $f = f_{\text{KL}}$, as $L \to \infty$, we recover the usual Fisher information $I_f^{\Gamma}(P\|Q) = E_P\left[|\nabla \log\left(\frac{p}{q}\right)|^2\right]$ which is used to prove convergence to the equilibrium state for the Fokker-Planck equation 25.

Functional inequalities, such as the classical Poincaré and the Logarithmic Sobolev-type inequalities, and many generalizations thereof see Markowich & Villani (2000); Otto & Villani (2000); Toscani & Villani (2000); Dolbeault et al. (2008); Wang (2005) are a powerful to prove convergence to equilibrium (e.g exponential or polynomial convergence), building on dissipation estimates such as Theorem B.1. For example, if for some $\lambda > 0$, a Sobolev inequality $D_f^{\Gamma}(P\|Q) \leq \frac{1}{\lambda} I_f^{\Gamma}(P\|Q)$ holds (true when $Q$ is sub-Gaussian), then we obtain exponential convergence to $Q$ for any $P_0$: $D_f^{\Gamma}(P_t\|Q) \leq e^{-\lambda t} D_f^{\Gamma}(P_0\|Q)$. There exits various results, e.g. Carrillo et al. (2006); Dolbeault et al. (2008); Markowich & Villani (2000); Wang (2008), reviewed in the Section B.3, B.4, on functional inequalities when $L = \infty$ which have been used to prove convergence to equilibrium (at exponential or polynomial rate) for Fokker-Planck and/or the porous media equation when the target distribution are Gaussian distribution, stretched exponential and Student-$t$ type distribution. The existence of functional inequalities for Lipschitz-regularized gradient flows is an open question.

## B.2 NUMERICAL ANALYSIS FOR LIPSCHITZ REGULARIZED PDES AND GENERATIVE PARTICLES

For a Lipschitz-regularized gradient flow (24), the transporter/discriminator representation (9) implies that the domain of dependence is determined by the velocity fields $\nabla \phi_t^{L,*}$ whose norm is bounded by the Lipschitz constant $L$. Therefore the domain of dependence of the solution is finite and is contained in a cone of slope $L$ that emanates from any point $(x, t)$ back to the time plane $t = 0$.

From a numerical analysis point of view, (10) is an explicit numerical scheme $\frac{p_{n+1}-p_n}{\Delta t} = \text{div}\left(p_n \nabla \phi_n^{L,*}(x)\right)$. For corresponding spatial discretization schemes there is an abundance of numerical methods which we can use to get some numerical analysis insight into our particle schemes (10). In particular, the Courant, Friedrichs, and Lewy (CFL) condition for stability of discrete schemes asserts that a numerical method can be convergent only if its numerical domain of dependence contains the true domain of dependence of the continuous PDE, LeVeque (2007). In our context, the CFL condition means $\sup_x |\nabla \phi_t^{L,*}(x)| \frac{\Delta t}{\Delta x} \leq C_{max}$ where $C_{max} = 1$ for such explicit schemes, LeVeque (2007). Clearly, the Lipschitz regularization enforces a CFL type condition with a learning rate $\Delta t$ proportional to the inverse of $L$. It remains an open question how to rigorously extend this CFL analysis to particle-based algorithms where the spatial discretization grid $\Delta x$ is known only implicitly as noted in Remark 4.1(b), see also related questions in Carrillo et al. (2017). Nevertheless, in our experiments we explore the inversely proportional relation between $L$ and $\Delta t$ suggested by the CFL analysis.

## B.3 FOKKER-PLANCK EQUATION, AND ITS CONVERGENCE TO EQUILIBRIUM STATE

**Generalized Fokker Planck as the gradient flow of $f$-divergences.** Let $p_t$ is the density of $P_t$. The associated gradient flow is given by the generalized Fokker-Planck equation

$$\partial_t p_t = \nabla \cdot \left(p_t \nabla \frac{\delta D_f(P\|Q)}{\delta P}(p_t)\right) = \nabla \cdot \left(p_t \nabla f'\left(\frac{p_t}{q}\right)\right) \tag{29}$$

**The Fokker Planck as gradient flow of KL.** When $f = f_{\mathrm{KL}}$, we obtain the known Fokker-Planck equation

$$\partial_t p_t - \Delta p_t + \nabla \cdot \left( p_t \frac{\nabla q}{q} \right) = 0.$$

### B.3.1 Exponential decay when $q \propto e^{-V}$ and $V$ is $\lambda$-convex

In this section for simplicity that the probability densities of both source and target distributions exist and are denoted by $p, q$. We consider the Cauchy problem of the Fokker-Planck equation given in Section B with

$$p(t = 0, \cdot) = p \geq 0 \text{ and } \int p = 1. \tag{30}$$

The next theorem in Markowich & Villani (2000) gives us the conditions that a probability measure satisfies in order to logarithmic Sobolev inequalities and consequently exponential decay.

**Theorem B.3.** *Let $q \in L^1(\mathbb{R}^d)$ and $V$ be $\lambda$-convex (i.e. $D^2 V(x) \geq \lambda I_d$ for all $x \in \mathbb{R}^d$), where $I_d$ is the identity matrix of dimension $d$. Then, $q$ satisfies a logarithmic Sobolev inequality with constant $\lambda$, i.e. $D_{\mathrm{KL}}(p\|q) \leq \frac{1}{2\lambda} I(p\|q)$, and the solution of the homogeneous Fokker-Planck equation goes to equilibrium in KL divergence, with a rate $e^{-2\lambda t}$ at least.*

Typical examples that satisfy the conditions of Theorem B.3 are

$$q(x) = \frac{e^{-|x|^\beta}}{\int e^{-|x|^\beta}}, \text{ for } x \in \mathbb{R}^d \quad \text{with} \qquad \beta \geq 2 \tag{31}$$

When $\beta = 2$, the target probability distribution with density $q$ is the Gaussian with variance $\sigma$ and zero mean, i.e.

$$q(x) = \frac{1}{(2\pi\sigma)^{d/2}} e^{-\frac{|x|^2}{2\sigma}},$$

(i.e. $V(x) = \frac{|x|^2}{2\sigma}$). By applying Theorem B.3, we get that for any initial probability distribution $P$ which is absolutely continuous with respect to $Q$,

$$D_{\mathrm{KL}}(p_t\|q) \leq D_{\mathrm{KL}}(p_0\|q) e^{-2t/\sigma} \tag{32}$$

where we have also used that the Stam-Gross Logarithmic Sobolev inequality, i.e. $D_{\mathrm{KL}}(p\|q) \leq \frac{\sigma}{2} I(p\|q)$, see formula (14) in Markowich & Villani (2000).

### B.3.2 Polynomial decay when $q \propto e^{-V}$ and $V$ is degenerately convex at infinity

We consider a potential $V \in W_{\mathrm{loc}}^{2,\infty}$ such that $\int q = 1$ and degenerately convex at infinity, i.e.

$$U(u) - a \leq V(u) \leq U(u) + b \tag{33}$$

where $a, b$ are nonnegative constants and $U$ is convex degenerate, i.e.

$$D^2 U(u) \geq c(1 + |u|)^{\beta-2}, \quad c > 0 \text{ and } \beta \in (0, 2) \tag{34}$$

Without loss of generality we assume that $U$ takes its unique minimum at 0. We further assume that for some $b, c, C_0 > 0$

$$\nabla V(u) \cdot u \geq c|u|^b - C_0 \tag{35}$$

A typical potential that satisfies (33), (34) and (35) is $V = |x|^\beta$ with $0 < \beta < 2$. Before we state the next theorem in Toscani & Villani (2000), we further define the following quantities

$$M_s(p) := \int p(x)(1 + |x|^2)^{s/2}, \quad s > 2 \text{ and } \delta := \frac{2 - \beta}{2(2 - \beta) + (s - 2)} \in (0, \frac{1}{2}) \tag{36}$$

**Theorem B.4.** *Let $V$ be a potential satisfying assumptions (33), (34) and (35). Let $p_0$ be a probability density such that $D_{\mathrm{KL}}(p_0\|q) < \infty$, $M_s(p_0) < \infty$ given in (36) for $s > 2$. Let also $\{p_t\}_{t\geq 0}$ be a (smooth) solution of the Fokker-Planck equation with potential $V$ and with initial datum $p_0$. Then, there is a constant $C$ depending on $D_{\mathrm{KL}}(p_0\|q), M_s(p_0)$. and $s$ such that for all $t > 0$,*

$$D_{\mathrm{KL}}(p_t\|e^{-V}) \leq \frac{C}{t^\kappa}, \quad \text{with} \quad \kappa = \frac{1 - 2\delta}{\delta} = \frac{s - 2}{2 - \beta}. \tag{37}$$

*where $\delta$ is given in (36).*

Note that as $\beta \to 2$, one recovers the usual logarithmic Sobolev inequality as discussed in Sect. B.3.1. We summarize the said examples in the following table.

Table 2: Rate of convergence to equilibrium state $q \propto e^{-V}$ in KL divergence

| Examples of $q \propto e^{-V}$ | Rate of convergence in KL divergence |
|---|---|
| $q \propto e^{-|x|^\beta}$, $\beta \geq 2$ | at least $e^{-2\lambda t}$ |
| Special case: $\mathcal{N}(0, \sigma)$ | at least $e^{-2\lambda t}$, with $\lambda = \frac{1}{\sigma}$ |
| $q \propto e^{-|x|^\beta}$, $0 < \beta < 2$ | $\mathcal{O}(t^{-\kappa})$, $\kappa$ as in (37) |

### B.4 WEIGHTED POROUS MEDIUM EQUATIONS AND THEIR CONVERGENCE TO EQUILIBRIUM STATE

#### B.4.1 WEIGHTED POROUS MEDIUM EQUATION

The gradient flow of $f$-divergences for likelihood ratio. One may rewrite (29) in terms of the likelihood ratio denoted by $h_t$ and defined as

$$h_t = \frac{dp_t}{dq} \tag{38}$$

By using the operator identity ($q$ being the multiplication operator by the function $q$), i.e.

$$\nabla q = q \left( \nabla + \nabla \log q \right)$$

we have that

$$\nabla \cdot p_t \nabla f' \left( \frac{p_t}{q} \right) = q(\nabla + \nabla \log q) h_t \nabla f'(h_t)$$

and thus we can rewrite (29) as

$$\partial_t h_t(x) = (\nabla + \nabla \log q) \cdot h_t \nabla f'(h_t) \tag{39}$$

Moreover if we denote $\nabla^*$ the adjoint of $\nabla$ on $L^2(q)$ we have $\nabla^* = -(\nabla + \nabla \log q)$ and thus (39) has the form

$$\partial_t h_t(x) = -\nabla^* h_t \nabla f'(h_t) \tag{40}$$

Let now $f_\alpha(x) = \frac{x^\alpha - 1}{\alpha(\alpha - 1)}$, we rewrite

$$h \nabla h^{\alpha - 1} = \frac{1}{\beta} \nabla v^\beta = v^{\beta - 1} \nabla r \implies v = h^{\alpha - 1} \text{ and } h = v^{\beta - 1} \implies \beta = \frac{\alpha}{\alpha - 1}$$

and thus we obtain

$$\partial_t h_t(x) = \frac{1}{\alpha} \left( \Delta + \nabla \log q \cdot \nabla \right) h_t^\alpha \tag{41}$$

for $t \geq 0$ and $x \in \mathbb{R}^d$ corresponding to a non-negative initial condition $h(x, 0) = h_0(x)$, $x \in \mathbb{R}^d$ is called weighted Porous Medium equation. For existence and uniqueness, see Dolbeault et al. (2008).

*Remark* B.5. The formula for $f_\alpha^*$ is given by

$$f_\alpha^*(y) = \begin{cases} \frac{(\alpha - 1)^{\frac{\alpha}{(\alpha - 1)}}}{\alpha} y^{\frac{\alpha}{(\alpha - 1)}} \mathbf{1}_{y > 0} + \frac{1}{\alpha(\alpha - 1)} & , \alpha > 1 \\ \infty \mathbf{1}_{y \geq 0} + \left( \frac{1}{\alpha(1 - \alpha)^{\frac{\alpha}{(1 - \alpha)}}} |y|^{-\frac{\alpha}{(1 - \alpha)}} \mathbf{1}_{y > 0} - \frac{1}{\alpha(1 - \alpha)} \right) \mathbf{1}_{y < 0} & , \alpha \in (0, 1) \end{cases} \tag{42}$$

*Remark* B.6. For completeness, we discuss a related gradient flow known as *granular media equation*. To be precise, the 2-Wasserstein gradient flow of $\mathcal{F}(p) = \frac{1}{2}\text{MMD}[p, q]^2$ where $\text{MMD}[p, q]$ is the Maximum mean discrepancy (MMD) Gretton et al. (2012). By recalling (2), MMD is defined as

$$\text{MMD}[p, q] = \sup_{g \in B_{\text{RKHS}}(0,1)} \{ E_Q[g] - E_P[g] \}$$

and its maximizer $\phi^*(z) = f_{q,p}(z) = \int k(x, z)q(x)dx - \int k(x, z)p(x)dx = k \star p(z) - k \star q(z)$ is called *witness function* between the probability densities $q$ and $p$. In fact, $g^*$ is the difference between the mean embeddings of $q$ and $p$ which finally makes MMD be re-written as the RKHS norm of the unnormalized $g^*$, i.e.

$$\text{MMD}[p, q] = \|\phi^*\|_{\mathcal{H}} \tag{43}$$

Then the gradient flow equation associated to $\mathcal{F}$ leads to the granular media equation, i.e

$$\partial_t p_t(x) = div \left( p \nabla \cdot (k \star p - k \star q) \right) \equiv div \left( p_t \nabla \phi_t^* \right) \tag{44}$$

B.4.2 FUNCTIONAL INEQUALITIES FOR THE WEIGHTED POROUS MEDIUM EQUATION

In this section, we apply Theorem 4.5 in Dolbeault et al. (2008) to Weighted Porous Medium for the likelihood ratio $h_t = \frac{p_t}{q}$ and we prove polynomial decay in KL and $\chi^2$-divergence. Before we state the result we first define the $L^r$-Poincaré inequality and $L^r$-logarithmic Sobolev inequality (see also Dolbeault et al. (2008)).

**Definition B.7.** Let $q$ be a probability measure on a Riemannian manifold $(M, g)$. Then the entropy is defined as follows: for any smooth function $f \in C^1(M)$

$$\mathbf{Ent}_q(f) := \int f \log\left(\frac{f}{\int f dq}\right) dq \tag{45}$$

while

$$\mathbf{Var}_q(f) := \int \left(f - \int f dq\right)^2 dq \tag{46}$$

**Definition B.8.** Let $q$ be a probability measure on a Riemannian manifold $(M, g)$. Let also $\nu$ be a positive measure on $(M, g)$. We assume that $q \in (0, 1]$. We say that $(q, \nu)$ satisfies $L^r$-Poincaré inequality with constant $C_P$ if and only if, for any nonnegative function $f \in C^1(M)$

$$\left[\mathbf{Var}_q(f^{2r})\right]^{1/r} \le C_P \int |\nabla f|^2 d\nu \tag{47}$$

We say that $(q, \nu)$ satisfies $L^r$-logarithmic Sobolev inequality with constant $C_{LS}$ if and only if, for any nonnegative function $f \in C^1(M)$

$$\left[\mathbf{Ent}_q(f^{2r})\right]^{1/r} \le C_{LS} \int |\nabla f|^2 d\nu \tag{48}$$

**Theorem B.9.** If $(q, q)$ satisfies a $L^{2/3}$-Poincaré Sobolev inequality, for some constant $C_P > 0$, then for any non-negative initial condition $h_0 \equiv \frac{p_0}{q} \in L^2(q)$, we have for every $t \ge 0$

$$\chi^2(p_t\|q) \le \left(\left[\chi^2(p_0\|q)\right]^{-1/2} + \frac{8}{9}C_P t\right)^{-2}. \tag{49}$$

*Reciprocally, if the above inequality is satisfied for any $g_0$, then $(q, q)$ satisfies a $L^{2/3}$-Poincaré Sobolev inequality with constant $C_P > 0$.*

**Theorem B.10.** Let $\alpha > 1$. If $(q, q)$ satisfies a $L^{1/\alpha}$-logarithmic Sobolev inequality, for some constant $C_{LS} > 0$, then for any non-negative initial condition $h_0$ such that $D_{KL}(p_0\|q) < \infty$, we have for every $t \ge 0$

$$D_{KL}(p_t\|q) \le \left([D_{KL}(p_0\|q)]^{1-\alpha} + \frac{4(\alpha-1)}{\alpha}C_{LS} t\right)^{-1/(\alpha-1)}. \tag{50}$$

*Reciprocally, if the above inequality is satisfied for any $g_0$, then $(q, q)$ satisfies a $L^{1/\alpha}$-logarithmic Sobolev inequality with constant $C_{LS} > 0$.*

Next we discuss two examples of probability distributions satisfy $L^r$-Poincaré inequality and $L^r$-logarithmic Sobolev inequality:

Let $r \in (0, 1]$ and $\beta \in [\frac{1}{2}, 1)$. The probability measure

$$dq = \frac{1}{2\Gamma\left(1 + \frac{1}{\beta}\right)} e^{-|x|^\beta} dx, \quad x \in \mathbb{R} \tag{51}$$

satisfies a $L^r$-Poincaré inequality and $L^r$-logarithmic Sobolev inequality.

Let $r \in [1/2, 1)$, then for $\beta > \frac{2r}{1-r}$ the probability measure

$$dq = \frac{\beta}{(1+|x|)^{1+\beta}} dx, \quad x \in \mathbb{R} \tag{52}$$

satisfies a $L^r$-Poincaré inequality and a $L^r$-logarithmic Sobolev inequality.

Table 3: Rate of convergence to equilibrium state $q \propto e^{-V}$ in $\chi^2$-divergence

| Examples of $q \propto e^{-V}$ | Rate of convergence in $\chi^2$ divergence |
|---|---|
| $q = \frac{e^{-|x|^\beta}}{2\Gamma\left(1+\frac{1}{\beta}\right)}$, $\;\; 0 < r \leq 1$, $\; 1/2 \leq \beta < 1$ | given in (49) |
| $q = \frac{\beta}{(1+|x|)^{1+\beta}}$, $\;\; 1/2 \leq r < 1$, $\; \beta \geq \frac{2r}{1-r}$ | given in (49) |

## C  FIRST VARIATION OF REGULARIZED DIVERGENCES

In this section we prove the following theorem

**Theorem C.1.** *Assume $f$ is superlinear and strictly convex and $P, Q \in \mathcal{P}_1(\mathbb{R}^d)$.*

1. *For $x \notin \text{supp}(P) \cap \text{supp}(Q)$ define $\phi^{L,*}(y) = \sup_{x \in \text{supp}(Q)} \left\{ \phi^{L,*}(x) + L|x-y| \right\}$ then $\phi^{L,*}$ is Lipschitz continuous on $\mathbb{R}^d$.*

2. *$\phi^{L,*} = \sup\{h(x) : h \in \Gamma_L, \, h(y) = \phi^{L,*}(y), \text{for every } y \in \text{supp}(Q)\}$*

3. *Let $\rho$ be a signed measure of total mass 0 and let $\rho = \rho_+ - \rho_-$ where $\rho_\pm \in \mathcal{P}_1(K)$ are mutually singular. If $P + \epsilon\rho \in \mathcal{P}_1(K)$ for sufficiently small $|\epsilon|$ then $D_f^{\gamma_L}(P + \epsilon\rho\|Q)$ is differentiable at $\epsilon = 0$ and we have*

$$\lim_{\epsilon \to 0} \frac{1}{\epsilon} \left( D_f^{\Gamma_L}(P + \epsilon\rho\|Q) - D_f^{\Gamma_L}(P\|Q) \right) = \int \phi^{L,*} d\rho \, .$$

   *In other words we have*

$$\frac{\delta D_f^{\Gamma_L}(P\|Q)}{\delta P}(P) = \phi^{L,*}$$

*Proof.* The proof of 1. is straightforward by using the triangular inequality of norms. For 2., since $h \in \Gamma_L$, we have that $h(x) \leq h(y) + \|x - y\|$. This implies that for $y \in \text{supp}(Q)$ and $x \notin \text{supp}(Q)$, $h(x) \leq \inf_{y \in \text{supp}(Q)}\{h(y) + \|x - y\|\} = \inf_{y \in \text{supp}(Q)}\{\phi^{L,*}(y) + \|x - y\|\} = \phi^{L,*}(x)$. Since $\phi^{L,*}(y) \in \Gamma_L$, this concludes the proof. For 3., we use the variational formula (3) for $D_f^{\Gamma_L}(P + \epsilon\rho\|Q)$ where we suppose that $P + \epsilon\rho \in \mathcal{P}_1(\mathbb{R}^d)$.

$$
\begin{aligned}
D_f^{\Gamma_L}(P + \epsilon\rho\|Q) &= \sup_{\phi \in \Gamma_L} \left\{ E_{P+\epsilon\rho}[\phi] - \inf_{\nu \in \mathbb{R}}\{\nu + E_Q[f^*(\phi - \nu)]\} \right\} \\
&\geq \int \phi^{*,L} d(P + \epsilon\rho) - \inf_{\nu \in \mathbb{R}} \left\{ \nu + \int f^*(\phi^{*,L} - \nu)dQ \right\} \\
&= \epsilon \int \phi^{*,L} d\rho + D_f^{\Gamma_L}(P\|Q) \quad (53)
\end{aligned}
$$

Thus

$$\liminf_{\epsilon \to 0^+} \frac{1}{\epsilon} \left( D_f^{\Gamma_L}(P + \epsilon\rho\|Q) - D_f^{\Gamma_L}(P\|Q) \right) \geq \int \phi^{*,L} d\rho$$

For the other direction: Let us define $F(\epsilon) = D_f^{\Gamma_L}(P + \epsilon\rho\|Q)$. By Theorem 18 and 71 in Birrell et al. (2022a), $F(\epsilon)$ is convex, lower semicontinuous and finite on $[0, \epsilon_0]$. Due to the convexity of $F$, $F$ is differentiable on $(0, \epsilon_0)$ except for a countable number of points. Let $\epsilon \in (0, \epsilon_0)$ such that $F$ is differentiable and $\delta > 0$ small. Also, let $\phi_\epsilon^{*,L}$ be the optimizer of $D_f^{\Gamma_L}(P + \epsilon\rho\|Q)$ satisfying $\phi_\epsilon^{*,L}(0) = 0$ so that

$$D_f^{\Gamma_L}(P + \epsilon\rho\|Q) = \int \phi_\epsilon^{*,L} d(P + \epsilon\rho) - \inf_{\nu \in \mathbb{R}} \left\{ \nu + \int f^*(\phi_\epsilon^{*,L} - \nu)dQ \right\}$$

By using the same argument as before in the proof, we have that

$$D_f^{\Gamma_L}(P + (\epsilon + \delta)\rho\|Q) - D_f^{\Gamma_L}(P + \epsilon\rho\|Q) \geq \delta \int \phi_\epsilon^{*,L} d\rho \quad (54)$$

and

$$D_f^{\Gamma_L}(P + (\epsilon - \delta)\rho\|Q) - D_f^{\Gamma_L}(P + \epsilon\rho\|Q) \geq -\delta \int \phi_\epsilon^{*,L} d\rho \tag{55}$$

which gives us that

$$
\begin{aligned}
\int \phi_\epsilon^{*,L} d\rho &\leq \lim_{\delta \to 0} \frac{1}{\delta} \left( D_f^{\Gamma_L}(P + (\epsilon + \delta)\rho\|Q) - D_f^{\Gamma_L}(P + \epsilon\rho\|Q) \right) \\
&= F'(\epsilon) \\
&= \lim_{\delta \to 0} \frac{1}{\delta} \left( D_f^{\Gamma_L}(P + \epsilon\rho\|Q) - D_f^{\Gamma_L}(P + (\epsilon - \delta)\rho\|Q) \right) \\
&\leq \int \phi_\epsilon^{*,L} d\rho
\end{aligned}
\tag{56}
$$

Consequently,

$$F'(\epsilon) = \int \phi_\epsilon^{*,L} d\rho \tag{57}$$

Let $F'_+(0) = \lim_{\epsilon \to 0^+} \frac{1}{\epsilon}(F(\epsilon) - F(0))$. By convexity, for any sequence $\{\epsilon_n\}_{n \in \mathbb{N}}$ such that $\epsilon_0 > \epsilon_n \downarrow 0$, we have

$$F'_+(0) = \lim_{n \to \infty} F'(\epsilon_n) = \lim_{n \to \infty} \int \phi_{\epsilon_n}^{*,L} d\rho$$

By applying the Arzelá-Ascoli to $\phi_{\epsilon_n}^{*,L}$, and then doing a diagonalization argument, there exists a subsequence of $\{n_k\}_{k \geq 0} \subset \{n\}_{n \geq 0}$, such that $\phi_{\epsilon_{n_k}}^{*,L}$ converges pointwise to a function $\phi_0^{*,L} \in \text{Lip}^L(\mathbb{R}^d)$. For simplicity, from now on we denote $n$ the convergent subsequence.

At this point, we recall that for any $\epsilon \in (0, \epsilon_0)$, $\phi_\epsilon^{*,L}(0) = 0$. For any $x$, $|\phi_\epsilon^{*,L}(x) - \phi_\epsilon^{*,L}(0)| \leq L\|x\|_d$ which implies that

$$|\phi_\epsilon^{*,L}(x)| \leq L\|x\|_d$$

Thus by the dominated convergence theorem

$$F'_+(0) = \lim_{n \to \infty} \int \phi_{\epsilon_n}^{*,L} d\rho = \int \phi_0^* d\rho$$

By the lower semicontinuity of $D_f^{\Gamma_L}(\cdot\|Q)$, we have

$$
\begin{aligned}
D_f^{\Gamma_L}(P\|Q) &\leq \liminf_{n \to \infty} D_f^{\Gamma_L}(P + \epsilon_n\rho\|Q) \\
&= \liminf_{n \to \infty} \left\{ E_{P+\epsilon_n\rho}[\phi_{\epsilon_n}^{*,L}] - \inf_{\nu \in \mathbb{R}} \{\nu + E_Q[f^*(\phi_{\epsilon_n}^{*,L} - \nu)]\} \right\} \\
&= \liminf_{n \to \infty} E_{P+\epsilon_n\rho}[\phi_{\epsilon_n}^{*,L}] - \limsup_{n \to \infty} \inf_{\nu \in \mathbb{R}} \{\nu + E_Q[f^*(\phi_{\epsilon_n}^{*,L} - \nu)]\} \\
&\leq E_P[\phi_0^{*,L}] - \inf_{\nu \in \mathbb{R}} \{\nu + E_Q[f^*(\phi_0^{*,L} - \nu)]\} \\
&\leq D_f^{\Gamma_L}(P\|Q)
\end{aligned}
\tag{58}
$$

where for the second inequality we use the dominated convergence theorem, (57) and that by Fatou's lemma

$$\limsup_{n \to \infty} \int \phi_{\epsilon_n}^{*,L} dQ \geq \liminf_{n \to \infty} \int \phi_{\epsilon_n}^{*,L} dQ \geq \int \phi_0^{*,L} dQ$$

Since both sides of the inequality coincide, $\phi_0^{*,L}$ must be the optimizer. By Theorem 3.1, part 1. and Theorem C.1, part 2., we have that $\phi_0^{*,L}(x) \leq \phi^{*,L}$ for all $x$. Thus

$$F'_+(0) = \int \phi_0^* d\rho \leq \int \phi^* d\rho.$$

which concludes the proof. $\qquad \square$

# D DETAILS ON LATENT GENERATIVE PARTICLES

## D.1 GENERALIZATION OF ENCODER AND DECODER FUNCTIONS

In the Section 5, we built Theorem 5.1 based on an encoder map $\mathcal{E} : \mathcal{Y} \to \mathcal{Z}$ (for instance $\mathcal{Y} = \mathbb{R}^d$, $\mathcal{Z} \subset \mathbb{R}^{d'}$, $d' << d$) and a decoder map $\mathcal{D} : \mathcal{Z} \to \mathcal{Y}$. For the generalization purpose, we identify these mappings as dirac kernels $K_{\mathcal{E}}(y, dz) = \delta_{\mathcal{E}(y)}(dz)$ and $K_{\mathcal{D}} = \delta_{\mathcal{D}(z)}(dy)$.

1. A pullback function induced by the kernel $K_{\mathcal{E}}$ is given as

$$K_{\mathcal{E}}[f](y) := \int f(z') K_{\mathcal{E}}(y, dz') = f(\mathcal{E}(y)) \tag{59}$$

for $f \in \mathcal{M}_b(\mathcal{Z}), y \in \mathcal{Y}$

2. A push-forward measure induced by the kernel $K_{\mathcal{E}}$ maps $\mathcal{P}(\mathcal{Y}) \to \mathcal{P}(\mathcal{Z})$ and defines a push-forward measure which is given as

$$K_{\mathcal{E}}[P](B) := \int K_{\mathcal{E}}(y, B) P(dy) = P \circ \mathcal{E}^{-1}(B) \tag{60}$$

for $P \in \mathcal{P}(\mathcal{Y})$ and a $\mathcal{Z}$-measurable set $B$.

Likewise, the kernel $K_{\mathcal{D}}$ induces a pullback function and a push-forward measure in the opposite direction. In the previous formulation, the $Q^{\mathcal{Y}}$-perfect encoding property $\mathcal{D}_{\#}\mathcal{E}_{\#}Q^{\mathcal{Y}} = Q^{\mathcal{Y}}$ can be rewritten as $Q^{\mathcal{Y}} = K_{\mathcal{D}}[K_{\mathcal{E}}[Q^{\mathcal{Y}}]]$. Given any $P^{\mathcal{Y}} \in \mathcal{P}(\mathcal{Y})$, we have the latent probability measure $P^{\mathcal{Z}} = K_{\mathcal{E}}[P^{\mathcal{Y}}] \in \mathcal{P}(\mathcal{Z})$ and the reconstructed probability measure $\tilde{P^{\mathcal{Y}}} = K_{\mathcal{D}}[K_{\mathcal{E}}[P^{\mathcal{Y}}]] \in \mathcal{P}(\mathcal{D}(\mathcal{Z}))$ where $\mathcal{D}(\mathcal{Z}) \subset \mathcal{Y} = \mathbb{R}^d$. In general, $\tilde{P^{\mathcal{Y}}} \neq P^{\mathcal{Y}}$.

Transition probability kernels are defined in the form of conditional distributions: $K_p(y, dz) = p(dz|y)$ from $\mathcal{Y}$ to $\mathcal{Z}$ and $K_q(z, dy) = q(dy|z)$ from $\mathcal{Z}$ to $\mathcal{Y}$. The kernel-induced pullback functions $K_p[f](y) = \int f(z') p(dz'|y) = \mathbb{E}_{Z|Y=y \sim p(dz|y)}[f(Z)|Y = y]$ or $K_q[g](z) = \int f(y') q(dy'|z) = \mathbb{E}_{Y|Z=z \sim q(dy|z)}[g(Y)|Z = z]$ are interpreted as conditional expectations. In addition, the kernels induce push forward measures $P^{\mathcal{Z}}(dz) = p(dz|y)P^{\mathcal{Y}}(dy)$ for $P^{\mathcal{Y}} \in \mathcal{P}(\mathbb{R}^d)$ or $R^{\mathcal{Y}}(dy) = q(dy|z)R^{\mathcal{Z}}(dy)$ for $R^{\mathcal{Z}} \in \mathcal{P}(\mathbb{R}^{d'})$. For the $Q$-perfect encoding property, we require these kernels to satisfy $dQ^{\mathcal{Y}}(dy) = q(dy|z)p(dz|y)dQ^{\mathcal{Y}}(dy)$.

## D.2 DATA PROCESSING INEQUALITY AND PROOF OF THEOREM 5.1

The proof of Theorem 5.1 is a consequence of a, new, tighter data processing inequality derived in Birrell et al. (2022a) that involves both transformations of probabilities and discriminator spaces $\Gamma$.

**Theorem D.1 (Data processing inequality for $(f, \Gamma)$ -divergences).** *Given a real valued convex function $f$, $P, Q \in \mathcal{P}(\Omega)$, and a probability kernel $K$ from $(\Omega, \mathcal{M})$ to $(N, \mathcal{N})$, if $\Gamma \subset \mathcal{N}$ is nonempty, then*

$$D_f^{\Gamma}(K[P] \| K[Q]) \leq D_f^{K[\Gamma]}(P \| Q). \tag{61}$$

*Proof.* From the variational formulation of divergences, we have

$$D_f^{\Gamma}(K[P] \| K[Q]) = \sup_{\phi \in \Gamma, \nu \in \mathbb{R}} \int \int (\phi(y) - \nu) K(x, dy) P(dx) - \int \int f^*(\phi(y) - \nu) K(x, dy) Q(dx). \tag{62}$$

Since $f^*$ is convex, Jensen's inequality gives

$$\int f^*(\phi(y) - \nu) K(x, dy) \geq f^* \left( \int (\phi(y) - \nu) K(x, dy) \right) \tag{63}$$

for all $x \in \Omega$. Hence,

$$D_f^{\Gamma}(K[P] \| K[Q]) \leq \sup_{\phi \in \Gamma, \nu \in \mathbb{R}} \mathbb{E}_P[K[\phi] - \nu] - \mathbb{E}_Q[f^*(K[\phi] - \nu)] = D_f^{K[\Gamma]}(P \| Q). \tag{64}$$

$\square$

Now we state and prove the generalized version of the Theorem 5.1.

**Theorem D.2.** *Suppose that*

1. **Perfect encoding.** *For $Q^{\mathcal{Y}}$ the encoder $\mathcal{E}$ and decoder $\mathcal{D}$ are such that $K_{\mathcal{D}}[K_{\mathcal{E}}[Q^{\mathcal{Y}}]] = Q^{\mathcal{Y}}$.*

2. *$K_{\mathcal{D}}[\Gamma_{\mathcal{Y}}] \subset \Gamma_{\mathcal{Z}}$. The pullback functions induced by the decoder kernel is included in the real function space.*

*Then, for any $P^{\mathcal{Z}} \in \mathcal{P}(\mathbb{R}^{d'})$ we have*

$$D_f^{\Gamma_{\mathcal{Y}}}(K_{\mathcal{D}}[P^{\mathcal{Z}}]\|Q^{\mathcal{Y}}) \leq D_f^{\Gamma_{\mathcal{Z}}}(P^{\mathcal{Z}}\|K_{\mathcal{E}}[Q^{\mathcal{Y}}]). \tag{65}$$

*Proof.* Since the encoder $\mathcal{E}$ and the decoder $\mathcal{D}$ perfectly reconstruct $Q^{\mathcal{Y}}$,

$$D_f^{\Gamma_{\mathcal{Y}}}(K_{\mathcal{D}}[P^{\mathcal{Z}}]\|Q^{\mathcal{Y}}) = D_f^{\Gamma_{\mathcal{Y}}}(K_{\mathcal{D}}[P^{\mathcal{Z}}]\|K_{\mathcal{D}}[K_{\mathcal{E}}[Q^{\mathcal{Y}}]]).$$

From data processing inequality,

$$D_f^{\Gamma_{\mathcal{Y}}}(K_{\mathcal{D}}[P^{\mathcal{Z}}]\|K_{\mathcal{D}}[K_{\mathcal{E}}[Q^{\mathcal{Y}}]]) \leq D_f^{K_{\mathcal{D}}[\Gamma_{\mathcal{Y}}]}(P^{\mathcal{Z}}\|K_{\mathcal{E}}[Q^{\mathcal{Y}}]).$$

By the assumption that $K_{\mathcal{D}}[\Gamma_{\mathcal{Y}}] \subset \Gamma_{\mathcal{Z}}$,

$$D_f^{K_{\mathcal{D}}[\Gamma_{\mathcal{Y}}]}(P^{\mathcal{Z}}\|K_{\mathcal{E}}[Q^{\mathcal{Y}}]) \leq D_f^{\Gamma_{\mathcal{Z}}}(P^{\mathcal{Z}}\|K_{\mathcal{E}}[Q^{\mathcal{Y}}]).$$

$\square$

### D.3 Latent generative particles algorithm

---
**Algorithm 2:** Latent Lipschitz regularized generative particles algorithm

---
**Require:** $f$ defined in (2) and its Legendre conjugate $f^*$, $L$: Lipschitz constant, $\nu$: scalar parameter for optimizing $f$ divergence, $T$: number of updates for the particles, $\gamma$: time step size, $N$: number of particles

**Require:** $W = \{W^l\}_{l=1}^D$: parameters for the neural network $\phi : \mathbb{R}^{d'} \to \mathbb{R}$, $D$: depth of the neural network, $\delta$: learning rate of the neural network, $T_{\text{NN}}$: number of updates for the neural network.

**Require:** $\mathcal{E} : \mathbb{R}^d \to \mathbb{R}^{d'}$: trained encoder, $\mathcal{D} : \mathbb{R}^{d'} \to \mathbb{R}^d$: trained decoder.

**Result:** $\{Y_T^{(i)}\}_{i=1}^N$

1 Sample $\{\bar{X}^{(i)} = \mathcal{E}(X^{(i)}) \in \mathbb{R}^{d'}\}_{i=1}^N$ where $\{X^{(i)}\} \sim Q$ is a batch from the real data

2 Sample $\{\bar{Y}_0^{(i)} = \mathcal{E}(Y^{(i)}) \in \mathbb{R}^{d'}\}_{i=1}^N$ where $\{Y^{(i)}\} \sim P_0 = P$ is a batch of prior samples

3 Apply Lipschitz regularized generative particles algorithm 1 on $\bar{X}^{(i)}$ and $\bar{Y}_0^{(i)}$

4 Reconstruct $Y_T^{(i)} = \mathcal{D}(\bar{Y}_T^{(i)})$

---

## E Experimental setting

**Neural network architectures.** We use the discriminator $\phi$ (compared to GAN setting) which is implemented using a neural network. In Table 4 we provide the architecture of the neural networks used to produce the experimental results. The Lipschitz constraint on $\phi$ is implemented by spectral normalization (the weight matrix in each layer of the $D$ layers in total has spectral norm $\|W^l\|_2 = L^{1/D}$) which is interpreted as a hard constraint. Imposing a gradient penalty term in the loss (soft constraint) was tried, but there were some problems: required additional tuning for the initial weight scales, and did not successfully restrict particle speeds bounded by $L$. Therefore we keep imposing the hard constraint in the entire paper.

**Data sets and important parameters.** See Table 5. More details can be found in Supplementary material README.md.

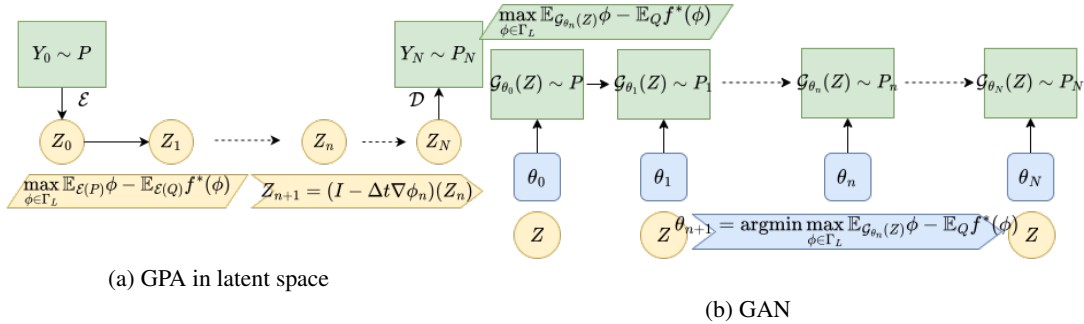

(a) GPA in latent space

(b) GAN

Figure 4: Workflow of different generative models. green: real space, yellow: latent space, blue: parameter space

| CNN Discriminator |
|---|
| $5 \times 5$ Conv SN, $2 \times 2$ stride $(1 \to ch_1)$ |
| leaky ReLU |
| $5 \times 5$ Conv SN, $2 \times 2$ stride $(ch_1 \to ch_2)$ |
| leaky ReLU |
| $5 \times 5$ Conv SN, $2 \times 2$ stride $(ch_2 \to ch_3)$ |
| leaky ReLU |
| Flatten with dimension $\ell_3$ |
| $W^4 \in \mathbb{R}^{\ell_3 \times d}$ with SN, $b^4 \in \mathbb{R}^d$ |
| ReLU |
| $W^5 \in \mathbb{R}^{d \times 1}$ with SN, $b^5 \in \mathbb{R}$ |
| Linear |

(a) Image data (MNIST, CIFAR10)

| FNN Discriminator |
|---|
| $W^1 \in \mathbb{R}^{d \times \ell_1}$ with SN, $b^1 \in \mathbb{R}^{\ell_1}$ |
| ReLU |
| $W^2 \in \mathbb{R}^{\ell_1 \times \ell_2}$ with SN, $b^2 \in \mathbb{R}^{\ell_2}$ |
| ReLU |
| $W^3 \in \mathbb{R}^{\ell_2 \times \ell_3}$ with SN, $b^3 \in \mathbb{R}^{\ell_3}$ |
| ReLU |
| $W^4 \in \mathbb{R}^{\ell_3 \times 1}$ with SN, $b^4 \in \mathbb{R}$ |
| Linear |

(b) Low dimensional data with dimension $d$

Table 4: Neural network architectures of the discriminator $\phi : \mathbb{R}^d \to \mathbb{R}$

**Computational resources.** Low dimensional examples are computed in the tensorflow-CPU environment: *tensorflow-gpu=2.8.0 with CPU model Intel(R) Core(TM) i5-10210U CPU @ 1.60GHz ~ 2.11 GHz.* Image generation examples are computed in the tensorflow-GPU environment: *tensorflow-gpu=2.7.0 with 1-GPU model Tesla K80 in Google cloud platform.*

## F  ADDITIONAL EXPERIMENTS

### F.1  COMPARISON WITH OTHER GENERATIVE DYNAMICS

We compare the GPA and other generative dynamics such as RKHS-based methods (Figure 5) and score-based methods by examples of 2D Mixture of Gaussians (Figure 6).

### F.2  ADDITIONAL LATENT GENERATIVE PARTICLES EXAMPLE

We applied $(f_{\text{KL}}, \Gamma_1)$ generative particle algorithm in the latent space and reconstructed to the high dimensional image data. In the high dimensional space, we first sampled initial particles from the *logistic distribution* and target data in $[0, 1]^{28 \times 28}$ for MNIST and in $[0, 1]^{32 \times 32 \times 3}$ for CIFAR10. For each of MNIST and CIFAR10, autoencoder with 128d latent dimension were trained. Then GPA was done in the 128d latent spaces. The number of training samples are $N = 200, 2000$.

### F.3  MICROARRAY GENE EXPRESSION DATA

The flexibility on the choice of source distributions and the small sample size regime enable our generative particles algorithm to be used for medical data-processing purpose. In addition, using

| Dataset | $f$ | $L$ | data parameter | NN structure | learning rate | $\Delta t$ | $N_Q$ |
|---|---|---|---|---|---|---|---|
| MNIST | KL | 1 | conditioned | CNN $(128, 128, 128)$ | 0.001 | 0.5 | 200, 2K |
| Gene data | KL | 1 | $d = 2, 5, 10$ $d = 20, 50, 100$ $d = 200$ | FFN $(32, 32, 32)$ FFN $(64, 64, 64)$ FFN $(128, 128, 128)$ | 0.1 | 5.0 | 245 |
| 2D Student-t Mixture | $\alpha = 2$ | 1 | $\nu = 0.5$ | FFN $(32, 32, 32)$ | 0.005 | 1.0 | 500 |
| MNIST | KL | 1 | conditioned $d' = 128$ | FNN $(256, 512, 256)$ | 0.001 | 0.5 | 200, 2K |
| CIFAR10 | KL | 1 | conditioned $d' = 128$ | FNN $(256, 512, 256)$ | 0.001 | 0.1 | 200, 2K |
| 2D Gaussian Mixture 1 | KL | 1, 10, 100, None | $\sigma_Q = 0.5$ | FFN $(32, 32, 32)$ | 0.005 | 1.0 | 200 |
| 2D Gaussian Mixture 2 | KL | 1 | $\sigma_Q = 1.0$ | FFN $(32, 32, 32)$ | 0.005 | 0.5 | 500 |

Table 5: Data sets and important parameters

| Sample size | 200 | 2000 |
|---|---|---|
| $(f_{\mathrm{KL}}, \Gamma_1)$-GPA | 4571.98 | 5143.55 |
| $(f_{\mathrm{KL}}, \Gamma_1)$-GAN | 5603.55 | 1270.13 |
| Wasserstein-GAN | 5653.20 | 1879.18 |

(a) Final FID for MNIST: GPA and GANs.

| Sample size | 200 | 2000 |
|---|---|---|
| MNIST | 1107.57 | 1048.84 |
| CIFAR10 | 87.34 | 75.10 |

(b) Final FID of MNIST and CIFAR10 (latent GPA).

Table 6: Conditional image generation performance summary. See Figure 1, 7.

latent generative particles scheme, we can effectively handle high-dimensional data, such as gene expression data, typically in the dimension of $5 \sim 6 \times 10^5$ depending on the probes in a significantly reduced dimensions. We suggest batch normalization/data merging as an application of our algorithm.

**Dataset.** We tested on publicly available gene expression data sets from Gene Expression Omnibus (https://www.ncbi.nlm.nih.gov/geo/):

- Breast cancer: Accession number GSE47109 (206 samples), GSE10843 (245 samples)

The former forms the source dataset, and the latter forms the target dataset. The source and the target data lie in the same dimensional space $\mathbb{R}^{54,675}$.

**Auto-encoder.** We applied PCA on the combined matrix for the source data and the target data which have been firstly normalized to mean zero and variance one. The normalized PCA can be interpreted as a linear auto-encoder. The PCA decoder is Lipschitz continuous with $L = \sqrt{d'}$. (Let $\mathbf{y} = \sum_{i=1}^{d'} z_i \mathbf{v}_i, \mathbf{y}' = \sum_{i=1}^{d'} z_i' \mathbf{v}_i \in \mathbb{R}^d$. The decoder $\mathcal{D}(\mathbf{z}) = \sum_{i=1}^{d'} z_i \mathbf{v}_i$ satisfies $\|\mathcal{D}(\mathbf{z}) - \mathcal{D}(\mathbf{z}')\| = \|\sum_{i=1}^{d'} (z_i - z_i') \mathbf{v}_i\| \leq \|\mathbf{z} - \mathbf{z}'\| \sqrt{\sum_{i=1}^{d'} \|\mathbf{v}_i\|^2} = \sqrt{d'}\|\mathbf{z} - \mathbf{z}'\|$ by Cauchy-Schwarz inequality. )

**Outputs** See Figures 8, 9 for the transported particles in the latent space for varying $d' = 2, 5, 10, 20, 50, 100, 200$ and the reconstructed space.

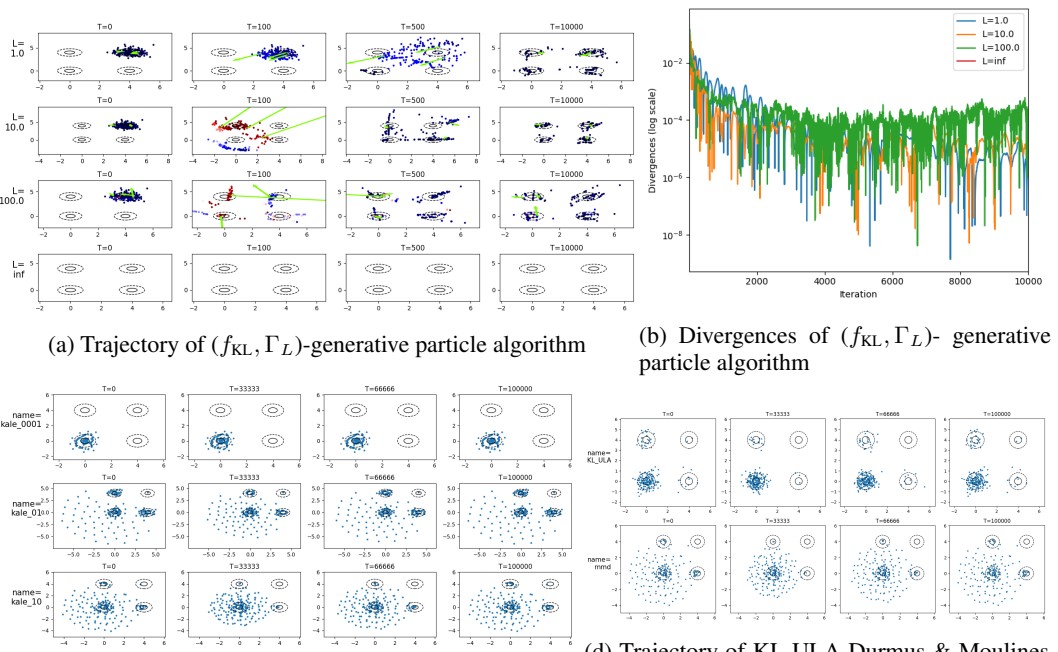

(a) Trajectory of $(f_{\text{KL}}, \Gamma_L)$-generative particle algorithm

(b) Divergences of $(f_{\text{KL}}, \Gamma_L)$- generative particle algorithm

(c) Trajectory of KALE flow Glaser et al. (2021) (Different (2017), MMD flow Arbel et al. (2019) regularizations)

(d) Trajectory of KL ULA Durmus & Moulines

Figure 5: **(2D Mixture of Gaussians 1) Comparison with RKHS based generative dynamics. (a)** $(f_{\text{KL}}, \Gamma_L)$-generative particles algorithm with different values for $L$. The particles are transported to the 4 wells faster as $L$ gets larger, however for large $L$ the algorithm become unstable ($L \geq 100$). Learning rates are chosen as $\gamma = 1.0, \delta = 0.005$. **(c)** KALE flow can be compared with $(f_{\text{KL}}, \Gamma_L)$-generative particles algorithm in the sense of being a different regularization technique. The KALE gradient flow regularizes the RKHS norm of $\phi^*$ while $(f_{\text{KL}}, \Gamma_L)$-generative particles algorithm regularizes the norm of $\nabla \phi^*$. The KALE flow Glaser et al. (2021) fails to capture the 4 wells in a reasonable amount of time. Here a Gaussian kernel with $\sigma = 0.5$ is chosen for the RKHS kernel. Learning rate is chosen as 0.001. **(d)** Bottom: MMD Arbel et al. (2019) gradient flow (without extra noise). A Gaussian kernel with $\sigma = 0.5$ is used for the RKHS. Top: For comparison KL gradient flow trained with the unadjusted Langevin algorithm (ULA) Durmus & Moulines (2017). The comparison with KL ULA and MMD flow suggests that the use of regularization enables the convergence without further techniques such as adding noise.

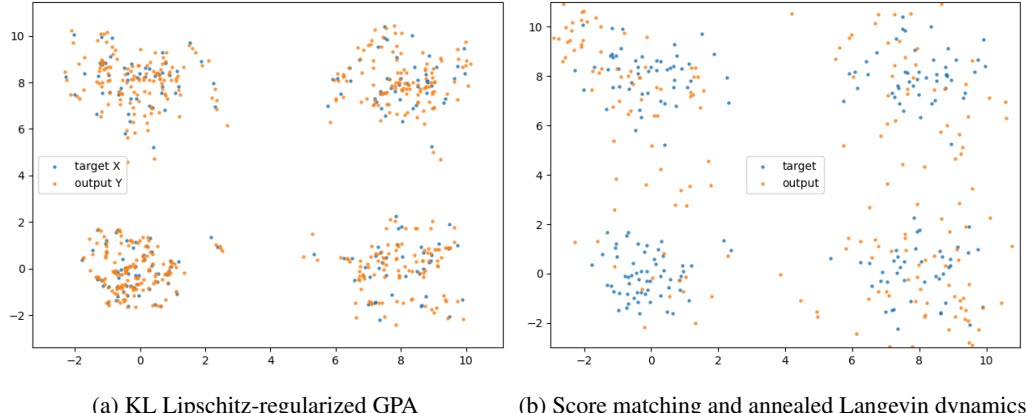

(a) KL Lipschitz-regularized GPA      (b) Score matching and annealed Langevin dynamics

Figure 6: **(2D Mixture of Gaussians)** $(f_{\text{KL}}, \Gamma_1)$**-GPA and the score based model (Noise conditional score network, NCSN).** 200 target samples from Mixture of Gaussians with $\sigma_Q = 1.0$ are provided to transport 500 particles which are uniformly distributed in the plotted region at time $t = 0$. Blue: target, Orange: output. **(a)** The choice of divergence $f_{\text{KL}}$ and propagation of particles through the $(f_{\text{KL}}, \Gamma_1)$-GPA captures the target. **(b)** shows learning a mixture of Gaussians is tractable using NCSN. Compare with a heavy-tailed target example in Figure 3.

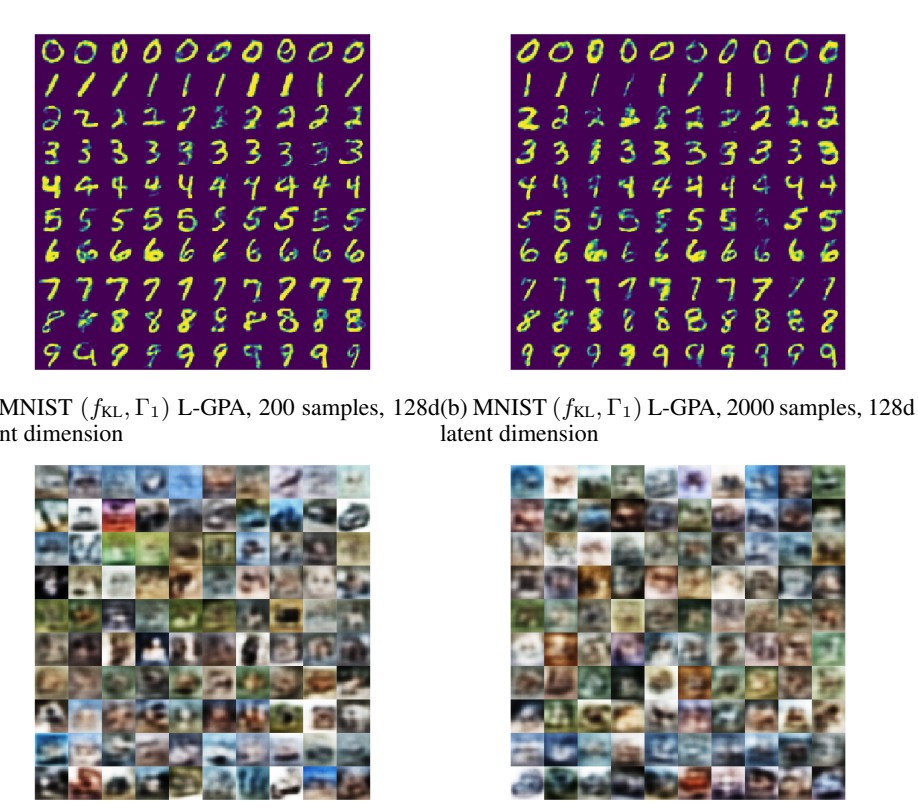

(a) MNIST $(f_{\text{KL}}, \Gamma_1)$ L-GPA, 200 samples, 128d latent dimension      (b) MNIST $(f_{\text{KL}}, \Gamma_1)$ L-GPA, 2000 samples, 128d latent dimension

(c) CIFAR10 $(f_{\text{KL}}, \Gamma_1)$ L-GPA, 200 samples, 128d latent dimension      (d) CIFAR10 $(f_{\text{KL}}, \Gamma_1)$ L-GPA, 2000 samples, 128d latent dimension

Figure 7: **GPA in the 128d latent space.** 200 samples are generated from 200, 2000 training data. See FID values in table 6.

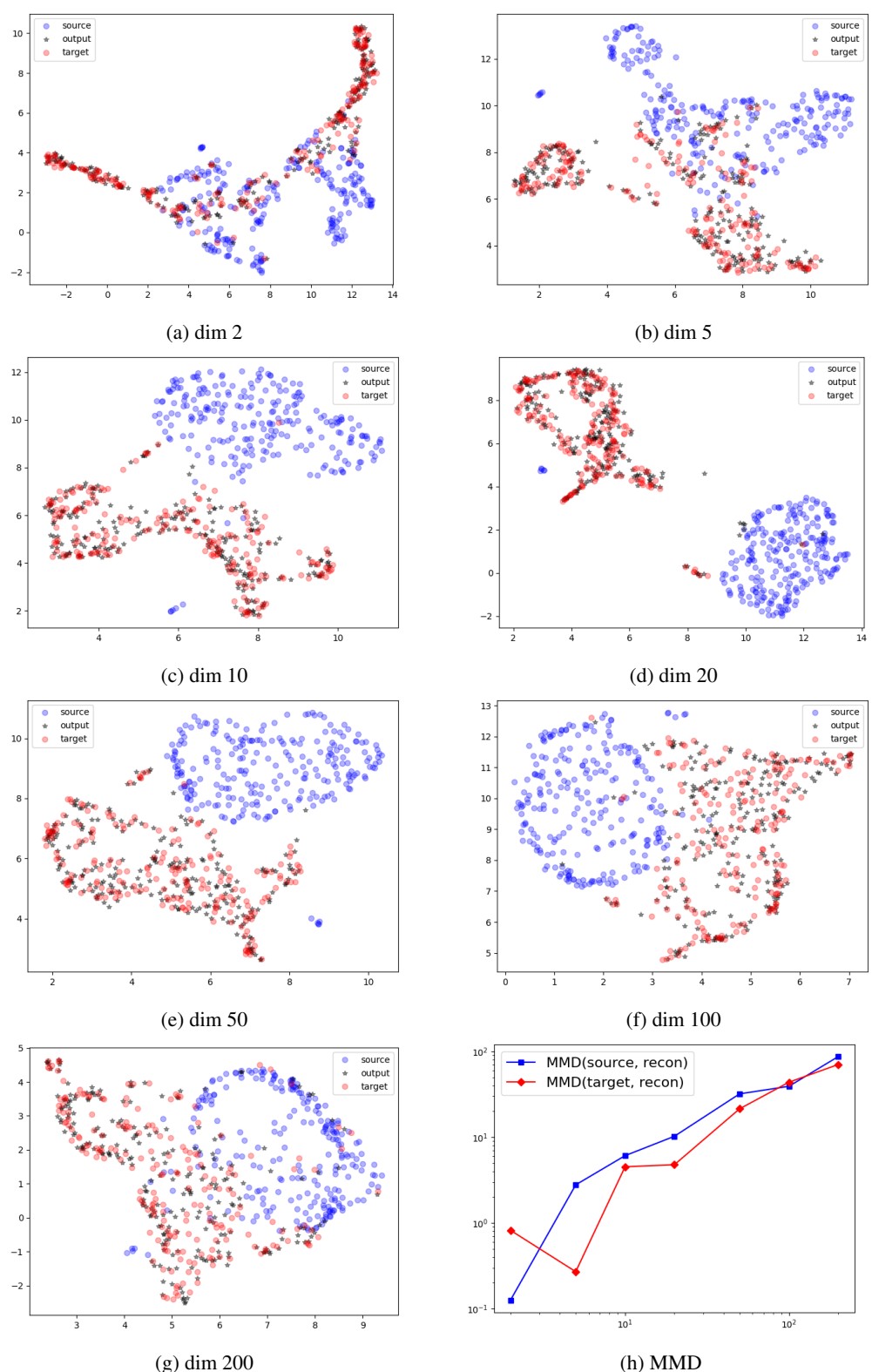

Figure 8: **(Gene expression data, BreastCancer) Latent samples. blue: source, red: target, black: transported.** (h) The distance between the latent distributions. **blue: MMD**$(P_0^{\mathcal{Z}}, P_T^{\mathcal{Z}})$, **red: MMD**$(Q^{\mathcal{Z}}, P_T^{\mathcal{Z}})$, **black: MMD**$(P_0^{\mathcal{Z}}, Q^{\mathcal{Z}})$ with $T = 25,000$.

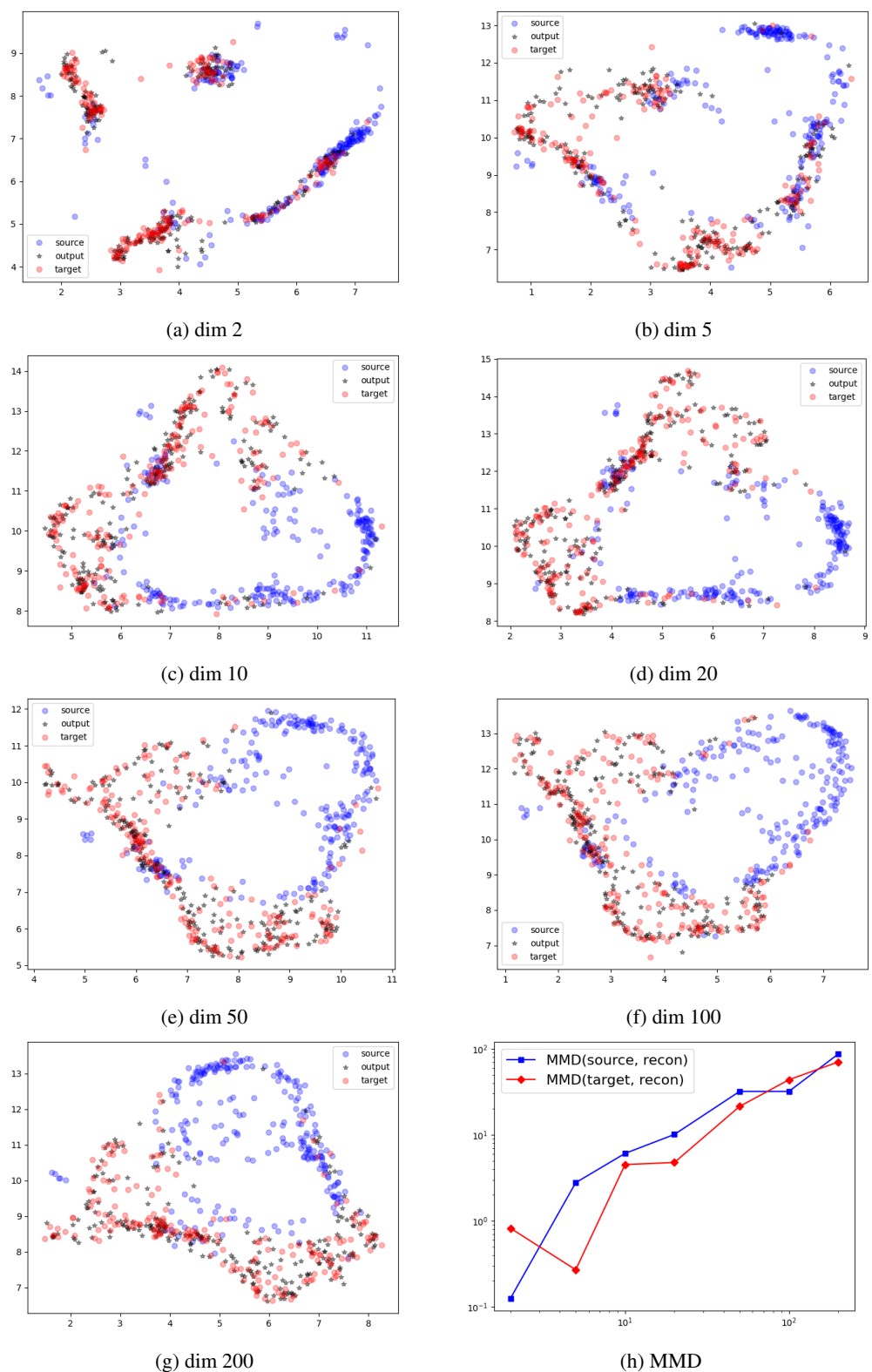

Figure 9: **(Gene expression data, BreastCancer) Reconstructed samples. blue: source, red: target, black: transported.** (h) The distance between the reconstructed distributions. **blue: MMD**$(P_0^{\mathcal{Y}}, P_T^{\mathcal{Y}})$**, red: MMD**$(Q^{\mathcal{Y}}, P_T^{\mathcal{Y}})$**, black: MMD**$(P_0^{\mathcal{Y}}, Q^{\mathcal{Y}})$ with $T = 25,000$.

