# OpenReview forum: "Lipschitz regularized gradient flows and latent generative particles"
_ICLR.cc/2023/Conference — Submitted to ICLR 2023_

### Official Review · Reviewer_hEpA · 2022-10-16

**Confidence:** 3
**Correctness:** 4
**Technical Novelty And Significance:** 3
**Empirical Novelty And Significance:** 2
**Recommendation:** 5

**Clarity, Quality, Novelty And Reproducibility:**

Clarity
-------
The paper is very clear. In particular, notation is tight, statements are supported well, and overall the paper is quite easy to read.

Quality
--------
The paper is of reasonably high quality. In particular, the theoretical developments are quite advanced and are executed well. The experiments may be a bit lackluster, however.

Novelty
---------
The paper is quite novel. It builds on prior work on gradient flows and f-divergences (for GANs) but offers a unique perspective. In particular, it should be noted that the central flow constructions are entirely novel and serve as a helpful bridge between the two (somewhat disconnected) communities.

Reproducibility
-----------------
The authors include a reproducibility statement which describes all relevant information.

**Strength And Weaknesses:**

Strengths
-----------
* The method is very nicely developed with a good interplay between f divergences, lipschitz regularization, and gradient flows. By using these three perspectives, it is able to introduce a unified perspective and build a real technical model.
* The presentation is very clear and is able to present the mathematically technical details without it being overly cumbersome. In particular, I appreciated the usage of notation.
* Additionally, the overall mathematical construction does not seem to have any flaws.

Weaknesses
---------------
* The experiments are somewhat weak. First, the experiments only show results for MNIST and Gene Expression (student-t seems more like a proof of concept than an experiment). These datasets are pretty toy (MNIST is rather low dimensional and easy to solve with simpler methods, and the Gene Expression dataset was just introduced to show merging).
* Furthermore, for MNIST in particular, it is very disconcerting that, while the method does perform well in the low-data regime, increasing data from 200->2000 samples doesn't improve the method at all (either visually or numerically). This seems like a potentially big drawback for the purported applications. In particular, if the comparison is on the low data regime only, the authors should compare with other methods that also look at this.
* The MNIST results also should be compared with modern methods. In particular, Wasserstein GANs are around 4 years old, and modern methods such as diffusion.
* There should be a connection with continuous diffusion models (in particular diffusion also has a similar connection with PDEs like the Fokker-Planck Equation) that the authors should elaborate on and include.
* It seems like the method might be pretty computationally expensive (e.g. Table 2 (b)). Could the authors comment more on the training time, as GANs are reasonably fast to train compared with most differential equation methods.

**Summary Of The Paper:**

This paper uses Lipschitz regularization (in the context of the dual for f-divergences) to construct gradient flows for sampling from target distributions. These distributions can be empirical (i.e. discrete), and the construction can be applied to build generative models. Experiments show improved performance for generative modeling in low-data regimes.

**Summary Of The Review:**

Overall, I lean (slightly) to accept the paper. This is mostly due to the theoretical niceness of the paper (including it's many connections with gradient flows, f-divergences, and PDEs). What's stopping me from being fully supportive is the experimental section, which includes relatively mixed results on small toy datasets. However, given the theoretical nature of the work, I do believe that the technical developments outweigh experimental shortcomings.

I would also ask the authors to include a section on diffusion models and the connections therein if possible.

Update
--------
Several of my questions have gone unanswered, which is rather concerning. Furthermore, in the rebuttal, the authors referenced "engineering issues" or "full optimization of algorithms" when talking about toy results (MNIST) or algorithmic running time. Given that there still seems to be a lot of necessary changes to be made, I believe it is better to revise and resubmit the paper and have thus changed my score to leaning reject.

---

> ### Author Response · Authors · 2022-11-19
> **Full responses to the reviews are available in the Supplementary material Responses_to_reviewers.pdf.**
>
> # Weaknesses
> 1. The experiments are somewhat weak. First, the experiments only show results for MNIST and Gene Expression (student-t seems more like a proof of concept than an experiment). These datasets are pretty toy (MNIST is rather low dimensional and easy to solve with simpler methods, and the Gene Expression dataset was just introduced to show merging).
>     * Response : Of course this is true regarding MNIST. One of the reasons we considered  the gene expression data sets is that they are  a fundamental  tool for life sciences in general. Here we have focused primarily on examples that can give a first demonstration of the capabilities of the proposed algorithms, and  on what problems they are best suited for. We are currently working on upscaling our methods to much larger data sets.
>
> 2. Furthermore, for MNIST in particular, it is very disconcerting that, while the method does perform well in the low-data regime, increasing data from 200 to 2000 samples doesn't improve the method at all (either visually or numerically). This seems like a potentially big drawback for the purported applications. In particular, if the comparison is on the low data regime only, the authors should compare with other methods that also look at this.
>     * Response : Indeed we agree this is an issue, we are currently working on resolving the engineering aspects of our software for the specific imaging applications. The very low data regimes give promising results, especially in examples such as the gene expression data set, where unlike in generative modeling for imaging problems, sophisticated data augmentation methods seem harder to develop.
>
> 3. The MNIST results also should be compared with modern methods. In particular, Wasserstein GANs are around 4 years old, and modern methods such as diffusion.
>     * Response : Regarding WGAN, our goal here was to compare GPA to another Wasserstein-based generative model. We also compared GPAs  to IPM-based methods that were developed more recently, such as MMD gradient flows and KALE, see also Appendix. We have already included a comparison with score-based diffusion models such as SGMs  where annealing is necessary. In the context of that example we show that GPAs do not  need to be injected with noise. GPAs are also capable of generating  distributions  with heavy tails where SGMs fail, see the last example in Section 6.
>
> 4. There should be a connection with continuous diffusion models (in particular diffusion also has a similar connection with PDEs like the Fokker-Planck Equation) that the authors should elaborate on and include.
>     * Response : We have added related references in the Introduction and a new Section in the Appendix on additional discussion on related work including on score-based diffusion models.
>
> 5. It seems like the method might be pretty computationally expensive (e.g. Table 2 (b)). Could the authors comment more on the training time, as GANs are reasonably fast to train compared with most differential equation methods.
>     * Response : In this first paper we  focused on the conceptual development of the method and the necessary  mathematics, and we are currently working  on the full optimization of the algorithms.
>
> # Summary Of The Review
> Overall, I lean (slightly) to accept the paper. This is mostly due to the theoretical niceness of the paper (including it's many connections with gradient flows, f-divergences, and PDEs). __What's stopping me from being fully supportive is the experimental section, which includes relatively mixed results on small toy datasets. However, given the theoretical nature of the work, I do believe that the technical developments outweigh experimental shortcomings. I would also ask the authors to include a section on diffusion models and the connections therein if possible.__
> * Response : We appreciate all the suggestions, and indeed we have included a full discussion on continuous-type diffusion models.

---

### Official Review · Reviewer_VduJ · 2022-10-20

**Confidence:** 4
**Correctness:** 3
**Technical Novelty And Significance:** 2
**Empirical Novelty And Significance:** 2
**Recommendation:** 3

**Clarity, Quality, Novelty And Reproducibility:**

The writing of the paper can be much improved.

* Quite a lot of sentences are not precise and thus confusing. For example, just above Section 4, "This is in sharp contrast with the Fokker Planck equation". How is the finite speed of propagation related to FP equation?
* Theorem 2.1 (except the 4th point) seems somewhat irrelevant to the rest of the paper
* I found the Data Processing Inequality and the argument on the mobility concept in Section 5 confusing and very handwavy. Aside from applying some change of variable in (13)(14) I don't get what the selling point is in this section.
* The experiments section is very confusing. It feels like the authors try to pack all the numbers and figures in the main text without properly explaining most of them. I think it would be better to focus on two experiments with greater details in the main text and move the rest to the appendix.
* Reproducibility seems good.


**Strength And Weaknesses:**

## Strengths:
* The idea of facilitating Wasserstein gradient flow by optimizing a learnable neural network from which the Wasserstein gradient can be obtained is a very interesting one.
* The proposed algorithm is simple and works for all kinds of $f$-divergences.

## Weaknesses:
* I found the novelty of the present paper limited. To me, the main novelty is (6), although I think the derivation is more or less identical to that of Dupuis & Mao (2022) from the KL case.
* The application prospect of the proposed algorithm is questionable. Without using the autoencoder, to me, the proposed algorithm is just to subsample the target empirical distribution. I think there is a lot to be explored here. For instance, if we use a very small number of particles (compared to the number of samples in the target distribution), can we still represent the target distribution faithfully? If so can we use the proposed method as an alternative to k-means?
* I don't think comparing with GAN is reasonable. GAN is capable of generating endless streams of new samples, whereas for the proposed method you must fix the number of particles ahead of time (it is also unclear to me how many particles were used in the experiments). A good baseline to compare would be, say minimizing the Wasserstein-1 (or W2) distance between particles and the target distribution and comparing the resulting particles. As a result, I do not find the experimental results convincing.


**Summary Of The Paper:**

This work proposes a particle flow algorithm that approximates a target empirical distribution with particles. The suggested algorithm updates the particles using the Wasserstein gradient of Lipschitz-regularized $f$-divergences, which is shown to exhibit a variational formulation as a supremum over Lipschitz continuous functions. An additional autoencoder can be used to flow the particles in a latent space. Various experiments are done to demonstrate the performance of the proposed method compared to GANs.


**Summary Of The Review:**

Overall I don't think the paper has enough novelty or has justified its application prospect well enough. The writing could also use a lot of improvement. As a result, I'm leaning toward rejection.

---

> ### Author Response · Authors · 2022-11-19
> **Full responses to the reviews are available in the Supplementary material Responses_to_reviewers.pdf.**
>
> # Weaknesses
> 1. I found the novelty of the present paper limited. To me, the main novelty is (6), although I think the derivation is more or less identical to that of Dupuis \& Mao (2022) from the KL case.
>     * Response : The variational derivative is indeed a key tool, but that is only one of the ingredients needed in the proposed methods, which on the theory side also  include: gradient descent, interacting particles formulation, dissipation estimates for the gradient descent,  and a new DPI for auto-encoders  to guarantee performance when using latent space calculations.
>
> 2. The application prospect of the proposed algorithm is questionable. Without using the autoencoder, to me, the proposed algorithm is just to subsample the target empirical distribution. I think there is a lot to be explored here. For instance, if we use a very small number of particles (compared to the number of samples in the target distribution), can we still represent the target distribution faithfully? If so can we use the proposed method as an alternative to k-means?
>     * Response : Indeed so far the performance on low number of samples seems very good, largely due to the use of the $(f, \Gamma)$-divergence that allows to transport efficiently arbitrary empirical to distributions to target  empirical distributions (combining the good mass transport features of  Wasserstein and the mass redistribution features of  KL). We have experimented with  various number of particles relative to the samples in the target distribution but we have not yet explored carefully the regime suggested by the referee. Indeed this is an intriguing direction, we greatly appreciate the suggestion.
>
> 3. I don't think comparing with GAN is reasonable. GAN is capable of generating endless streams of new samples, whereas for the proposed method you must fix the number of particles ahead of time (it is also unclear to me how many particles were used in the experiments). A good baseline to compare would be, say minimizing the Wasserstein-1 (or W2) distance between particles and the target distribution and comparing the resulting particles. As a result, I do not find the experimental results convincing.
>     * Response : The similarities and differences with GANs are discussed in the Table in Section 5. However, we are not sure we completely understand the suggestion by the referee. The referee proposes to minimize Wasserstein distances from the target but we are not sure over what models the optimization should be done.
>
> # Clarity, Quality, Novelty And Reproducibility
> 1. Quite a lot of sentences are not precise and thus confusing. For example, just above Section 4, "This is in sharp contrast with the Fokker Planck equation". How is the finite speed of propagation related to FP equation?
>     * Response : Hopefully these ambiguities are all fixed now: we added more discussion in a new Background component in Section 3  and added  a new Section in the Appendix on comparisons with other methods, where the finite speed of propagation is discussed, see also the Appendix Section on numerical analysis aspects of  PDEs.
>
> 2. Theorem 2.1 (except the 4th point) seems somewhat irrelevant to the rest of the paper
>     * Response : Theorem 2.1 provides background for the proposed divergences, for instance demonstrates how they interpolate between f-divergences and Wasserstein, while retaining good properties of both due to the infimum convolution representation.
>
> 3. I found the Data Processing Inequality and the argument on the mobility concept in Section 5 confusing and very handwavy. Aside from applying some change of variable in (13)(14) I don't get what the selling point is in this section.
>     * Response : The  Data Processing Inequality for auto-encoders in Theorem 5.1  provides  an  error estimate for the approximation of the real space model by the one built on the latent space: in the spirit of  numerical analysis of finite elements, e.g. \cite{AINSWORTH1997_aposteriori} we can use the right-hand side of Theorem 5.1 as an _a posteriori_ numerical estimate
> to provide  computable performance guarantees, since the upper bound is in the tractable latent space.
>
> 4. The experiments section is very confusing. It feels like the authors try to pack all the numbers and figures in the main text without properly explaining most of them. I think it would be better to focus on two experiments with greater details in the main text and move the rest to the appendix.
>     * Response : We have completely rearranged the Section and followed the suggestions of the referee for a more clear and precise delivery of the findings. Remaining material was moved in the Appendix. We now present three experiments (1) An analysis of MNIST, (2) The merging of medical datastsets (a "small data" problem well suited to our algorithms), and (3) A  model with very heavy tails where diffusion models fails but suitable $f$ divergences provide efficient learning algorithms.

---

> > ### Comment · Reviewer_VduJ · 2022-11-23
> > **Response to authors**
> >
> > Thank you for your response. I think the clarity of the paper has much improved. However, my central question is unanswered, which is: what is the application aspect of the algorithm, if not only used for subsampling a given dataset? Since the proposed algorithm is not generative (it cannot generate new particles aside from the initial ones), to this end I don't think the paper has demonstrated enough comparison to alternative methods that can also "thin out" a dataset. To give two examples:
> > - k-means clustering is designed for this task, and I think it is capable of producing results like in Figure 3(a)
> > - You can also minimize the Wasserstein distance or MMD between a small set of particles and the data distribution (if it's too big you can use mini-batches) over the positions of the particles. The resulting set of particles also provides a summary of the bigger data distribution.
> > - Or even more naively, you can simply draw a random subset of the dataset uniform at random. Why not just do this?
> >
> > If the goal is just to thin out a dataset, it is unclear what the advantages are to solve the expensive particle flow.
> >
> > As such, I would like to keep my current score for now.

---

### Official Review · Reviewer_VqE2 · 2022-10-25

**Confidence:** 2
**Correctness:** 3
**Technical Novelty And Significance:** 3
**Empirical Novelty And Significance:** 2
**Recommendation:** 5

**Clarity, Quality, Novelty And Reproducibility:**

**Clarity:** The paper is fairly clear given the technical nature of the results. There are some areas that can be improved. In particular, there is a lack of motivation for consideration of Lipschitz regularized functionals over continuous and bounded functions. I can see similarities with W-GAN in this regard, but more discussion along these lines would be nice. Also, Figure 1 comparing the differences between GPA and GAN is difficult to read and is quite unclear.

**Novelty and significance:** To the reviewer’s knowledge, the use of this general Lipschitz regularized class of divergences for generative modeling appears novel. The technical tools, however, appear heavily inspired by Birrell et al., and it is not clear what are the new contributions on the theoretical side.

**Reproducibility:** The empirical results appear reproducible with the code provided by the authors. I have not run the code myself, however.

**General comments:**
- It is curious that the sample generations do not appear to improve with more examples. In fact, in the MNIST example, the FID score is slightly worse for 2000 samples versus 200 samples. Do the authors have a sense of why this is the case? The theory seems to suggest that having a perfect encoder decoder pairing is important. Is there a connection here to a lack of a perfect encoder decoder pairing? Also, why does it seem to be that the model performs well with few samples?


**Strength And Weaknesses:**

**Strengths:**
- The proposed methodology of the paper is mathematically grounded, with some nice results on the approximation side showcasing that the error between the two distributions can be quantified.
- The method appears to perform at its best, and fairly well in low-data regimes.

**Weaknesses:**
- The motivation for the approach could be strengthened. In particular, it is unclear what the benefits are in terms of utilizing this Lipschitz regularized formulation versus the original variational formulation over continuous and bounded functions. More motivation and discussion along these lines would improve the readability of the paper.
- Some experimental results show that the generative capabilities do not improve with more data. For example, the MNIST generation results do not appear to improve with more samples both visually and quantitatively (as given by the FID score).

**Summary Of The Paper:**

This paper considers a new class of f-divergences that incorporate Lipschitz continuous functions in the variational representation of the f-divergence. These new class of divergences interpolate between the 1-Wasserstein metric and f-divergence when the Lipschitz parameter of the function class varies from 0 to infinity. This class also allows one to define a gradient flow and transportation equation from one distribution to another. Using this theory, the authors introduce a scheme to learn a generative model of a data distribution by learning a particle flow in the latent space of an auto encoder. This structure leverages the fact that many datasets of interest have intrinsic low-dimensionality. Experimental results are showcased on small synthetic and toy datasets such as MNIST.

**Summary Of The Review:**

Overall, I think that there are interesting mathematical ideas presented in this work, and the experimental results showcase some potential in the applications of such ideas as well. However, there are certain points about the core contributions on the theoretical side and puzzling properties about the model that I would like the authors to comment on before increasing my score.

---

> ### Author Response · Authors · 2022-11-19
> **Full responses to the reviews are available in the Supplementary material Responses_to_reviewers.pdf.**
>
> # Weaknesses
> 1. The motivation for the approach could be strengthened. In particular, it is unclear what the benefits are in terms of utilizing this Lipschitz regularized formulation versus the original variational formulation over continuous and bounded functions. More motivation and discussion along these lines would improve the readability of the paper.
>     * Response : We have added a new  explanatory Background subsection in Section 3 to address these issues. We also added  a corresponding numerical experiment.
>
> 2. Some experimental results show that the generative capabilities do not improve with more data. For example, the MNIST generation results do not appear to improve with more samples both visually and quantitatively (as given by the FID score).
>     * Response : We agree this is an issue and  we are currently working on resolving the engineering aspects of our software for the specific imaging applications and in general on upscaling our methods. Here we have focused primarily on examples that can give a first demonstration of the capabilities of the proposed algorithms, and  on what problems they are best suited for, for instance the ability to learn from very few samples.
>
> # Clarity, Quality, Novelty And Reproducibility
> __Clarity.__ The paper is fairly clear given the technical nature of the results. There are some areas that can be improved. In particular, there is a lack of motivation for consideration of Lipschitz regularized functionals over continuous and bounded functions. I can see similarities with W-GAN in this regard, but more discussion along these lines would be nice. Also, Figure 1 comparing the differences between GPA and GAN is difficult to read and is quite unclear.
> * Response : To address these issues, we added more discussion under ``related work" in the Introduction, a new Background component in Section 3  and added  a new Section in the Appendix on comparisons with other methods. We also refer to the Table in Section 5 using the concept of mobility as means for comparing  with GANs.
>
> __Novelty.__ To the reviewer’s knowledge, the use of this general Lipschitz regularized class of divergences for generative modeling appears novel. The technical tools, however, appear heavily inspired by Birrell et al., and it is not clear what are the new contributions on the theoretical side.
> * Response : The main new elements compared to that paper are the new Gradient flows and associated particle dynamics; the use of latent spaces and "latent" particles; and,  the related Data Processing Inequality for auto-encoders in Theorem 5.1  that provide an _a posteriori_ error estimate for the approximation of the real space model by the one built on the latent space.
>
> __General comments.__ It is curious that the sample generations do not appear to improve with more examples. In fact, in the MNIST example, the FID score is slightly worse for 2000 samples versus 200 samples. Do the authors have a sense of why this is the case? The theory seems to suggest that having a perfect encoder decoder pairing is important. Is there a connection here to a lack of a perfect encoder decoder pairing? Also, why does it seem to be that the model performs well with few samples?
> * Response : Indeed these are both  good questions. For now GPA seems to perform really well in cases  with few available samples, as demonstrated in the experiments. From a theory perspective the performance on low number of samples seems very good due to the use of the $(f, \Gamma)$-divergence that allow to transport efficiently arbitrary empirical to  target  empirical distributions.
> We are currently working on resolving the issues related to  scaling up and optimizing our algorithms.

---

### Official Review · Reviewer_Ds4C · 2022-10-26

**Confidence:** 3
**Correctness:** 3
**Technical Novelty And Significance:** 3
**Empirical Novelty And Significance:** 3
**Recommendation:** 5

**Clarity, Quality, Novelty And Reproducibility:**

The paper is very clear, and has high quality. The idea of using restricted variational forms of f-divergences for gradient flow appears in the following paper. But the combination of using the Lipschitz constrained class and latent space is original.

J. Fan, Q. Zhang, A. Taghvaei, Y. Chen. "Variational Wasserstein gradient flow", ICML 2022

**Strength And Weaknesses:**

Strength:
- The idea of considering gradient flows for Lipschitz constrained f-divergences is new to me
- Also, considering the gradient flows in the latent space seems interesting and valuable
- Numerical experiments are complete and informative
- the comparison Table 2 is very nice, though I wished to see more discussion.

Weakness:
- Not enough motivation for constraining in terms of Lipschitz functions, since neural networks are not a good representative of all Lipschitz functions with a certain Lipschitz constant. Why not gradient-norm penalty?
- The claim that Lipschitz constant makes the particle system "stable" should be made more precise. Stability in what sense?
- The infinite-speed of diffusions is true, but diffusion is not the only way to implement Fokker-Plank. It can also be implemented deterministically using \nabla \log (density) which is Lipschitz under standard assumptions on density.
- Not in depth discussion of the numerical results, conclusions, and limitations.




**Summary Of The Paper:**

The paper studies Wasserstein gradient flows for a certain class of objective functionals. These functionals form are an approximation of the f-divergence defined using its variational form. In particular, the optimization domain in the variational formulation is restricted to the class of functions with bounded Lipschitz constants. As the bound on the Lipschitz constant grows, these approximations become exact. The gradient flow is constructed with the objective of generating new samples from a target distribution (similar to the objective in GAN). Another presented idea is to consider generating the gradient flows in the latent space with an encoded-decoder mechanism in order to take advantage of the low-dimensional structure present in the target distribution.

**Summary Of The Review:**

The paper is well written in general and contain original ideas. However, it needs more motivation for considering Lipschitz constrained class. Also, it discusses two very nice and independent ideas in one single paper. Gradient flows in the latent space is independent of Lipschitz constrained gradient flows. And it is not clear that which of these two ideas are responsible for the improved numerical results compared to GAN.

####
I thank the authors for their response. But I am still not convinced about the motivation for considering Lipschitz constrained since most gradient flows are Lipschitz under mild assumptions on the initial and target density. After reading other reviews I am changing the score to 5.

---

> ### Author Response · Authors · 2022-11-19
> **Full responses to the reviews are available in Supplementary material Response_to_reviewers.pdf.**
>
> # Weaknesses
> 1. Not enough motivation for constraining in terms of Lipschitz functions, since neural networks are not a good representative of all Lipschitz functions with a certain Lipschitz constant. Why not gradient-norm penalty?
>     * Response : We impose the Lipschitz constraint directly on the neural nets using spectral normalization, \cite{Miyato_spectral_norm}.
> But the referee is correct, we could have also used a soft constraint as in GP-WGANs \cite{GP-WGAN} or in Section 4 of \cite{birrell2022f}. We included some relevant details in Appendix E. "Neural network architectures" paragraph.
>
> 2. The claim that Lipschitz constant makes the particle system "stable" should be made more precise. Stability in what sense?
>     * Response : By that statement we mean:  (1) stability in the  numerical analysis sense in terms of CFL condition. This is discussed in some detail in the Appendix. Furthermore, (2) stability due  to  regularization of the KL divergence by the Wasserstein (see Theorem 2.1 part 2) that allows us to compare empirical distributions of data. We have added an explanatory  Background subsection in Section 3.
>
> 3. The infinite-speed of diffusions is true, but diffusion is not the only way to implement Fokker-Plank. It can also be implemented deterministically using $\nabla \log $ (density) which is Lipschitz under standard assumptions on density.
>     * Response : We included such a discussion on deterministic probabilistic flows in the new version of the paper, both in the main text  and in the appendix. However we want to point out that the Lipschitz condition mentioned by the referee is practically uncheckable in generative modeling (but can be true in sampling problems based on maximum principle arguments for parabolic PDE). Furthermore such regularity conditions can be checked only if there is a well-defined density. In our context we have the ability to work directly with the empirical distribution of the data.
>
> 4. Not in depth discussion of the numerical results, conclusions, and limitations.
>     * Response : We trimmed, reorganized and summarized  our findings  in Section 6 and added more discussion of results in the Appendix.
>
>
> # Clarity, Quality, Novelty And Reproducibility
> The paper is very clear, and has high quality. The idea of using restricted variational forms of f-divergences for gradient flow appears in the following paper. But the combination of using the Lipschitz constrained class and latent space is original.
> _J. Fan, Q. Zhang, A. Taghvaei, Y. Chen. "Variational Wasserstein gradient flow", ICML 2022_
>
>
> * Response : We would like to thank the referee for pointing out this highly relevant paper. Restricted variational forms of f-divergences have appeared in numerous papers in the literature. However the paper \cite{birrell2022f} provides a unifying approach that also encompasses previous methods, we refer to that paper for both literature discussion and new results.
>  Both the present  manuscript and the paper cited by the referee \cite{JKO_ICML2022} focus on a gradient flow perspective for their corresponding  divergences. However, the time-discretization of their respective  gradient flows is different. We use the forward-Euler discretization using the discriminator while in \cite{JKO_ICML2022} the  authors use the implicit JKO scheme. Therefore, in \cite{JKO_ICML2022}, just like in GANs, an additional  Neural Net needs to be introduced in their algorithm compared to ours. The connections between the two papers remain intriguing  because of the similarity between the 2-Wasserstein-based JKO and our 1-Wasserstein infimal convolution formula in Theorem 2.1.
>
> # Summary Of the Review
> The paper is well written in general and contain original ideas. However, it needs more motivation for considering Lipschitz constrained class. Also, it discusses two very nice and independent ideas in one single paper. Gradient flows in the latent space is independent of Lipschitz constrained gradient flows. And it is not clear that which of these two ideas are responsible for the improved numerical results compared to GAN
>
> * Response : We have  explored GPA with and without  latent spaces in the Examples. But definitely a latent space can be  of huge help, and in addition  we also have related theoretical guarantees in Theorem 5.1.

---

### Decision · Program_Chairs · 2023-01-20

**Decision:**

Reject

**Justification For Why Not Higher Score:**

Lack of motivation and novelty of the proposed method.

**Justification For Why Not Lower Score:**

N/A

**Metareview: Summary, Strengths And Weaknesses:**

The paper studies the Wasserstein gradient flow of some regularised version of f-divergences. These regularizations  are obtained by restricting the variational formulation of f-divergences to functions with bounded Lipschitz constant. The gradient flow allows transporting an initial distribution toward a target by the regularised divergence. The paper then proposes a latent generative model obtained by performing a particle flow in the latent space of an auto-encoder.

After discussion with the reviewers, it appears that the paper would benefit from additional work to address the concerns on novelty and motivation.

Novelty: It appears that the idea of constraining the Lipschitz constant in the variational formulation of f-divergences is not novel as pointed out by Reviewers Ds4C, VqE2 and VduJ. The response of the authors suggests that one of the main contributions are to consider the Wasserstein gradient flow of these functionals, which is rather limited given that it results from classical constructions of the Wasserstein gradient flows. Therefore, the paper would require major changes to focus more on the third contribution (gradient flow in latent space) and provide more context on the known results.

Motivation: the motivation for the proposed approach is limited, as pointed out by Reviewers VduJ in response to the authors. This could be improved by considering concrete applications and comparing the proposed approach with some known baselines (see the reviewer's comments).